# Deglacial release of petrogenic and permafrost carbon from the Canadian Arctic impacting the carbon cycle

Junjie Wu [1,6] ✉, Gesine Mollenhauer [1,2] ✉, Ruediger Stein [1,2,3] ✉, Peter Köhler [1], Jens Hefter [1], Kirsten Fahl [1], Hendrik Grotheer [1], Bingbing Wei [4] & Seung-Il Nam [5]

The changes in atmospheric $p$CO$_2$ provide evidence for the release of large amounts of ancient carbon during the last deglaciation. However, the sources and mechanisms that contributed to this process remain unresolved. Here, we present evidence for substantial ancient terrestrial carbon remobilization in the Canadian Arctic following the Laurentide Ice Sheet retreat. Glacial-retreat-induced physical erosion of bedrock has mobilized petrogenic carbon, as revealed by sedimentary records of radiocarbon dates and thermal maturity of organic carbon from the Canadian Beaufort Sea. Additionally, coastal erosion during the meltwater pulses 1a and 1b has remobilized pre-aged carbon from permafrost. Assuming extensive petrogenic organic carbon oxidation during the glacial retreat, a model-based assessment suggests that the combined processes have contributed 8 ppm to the deglacial CO$_2$ rise. Our findings suggest potentially positive climate feedback of ice-sheet retreat by accelerating terrestrial organic carbon remobilization and subsequent oxidation during the glacial-interglacial transition.

Identifying Earth system processes that have contributed to atmospheric $p$CO$_2$ variability since the Last Glacial Maximum (LGM) remains one of the grand challenges in paleoclimatic research. During the last deglaciation, atmospheric CO$_2$ concentrations rose by ~75 ppm while the $\Delta^{14}$C values of CO$_2$ declined[1–3]. Evidence from both data and models suggests that the $p$CO$_2$ increase is driven mainly by changes in Southern Ocean ventilation, making the region a source of ancient carbon to the atmosphere[4,5]. However, the oceanic CO$_2$ release alone cannot account for the full glacial/interglacial amplitude of CO$_2$, or the rapid rise at 16.4, 14.6, and 11.5 cal. kyr BP, or the stable carbon isotopic anomaly of atmospheric CO$_2$. Hence, terrestrial and marine carbon sources have been both invoked to explain the CO$_2$ variations[6,7].

Ice sheets have traditionally been regarded as an inert component in the carbon cycle and thus have not been considered to explain the CO$_2$ variations. Growing evidence of terrestrial carbon remobilization/oxidation related to ice-sheet retreat, however, has indicated the necessity of a careful re-evaluation of this paradigm[8,9]. One can also imagine that glacial retreat directly leads to exposure to the atmosphere of organic carbon (OC) that has been isolated for many millennia and the oxidation of which may be important for the OC-rich regions. More recently, ice-sheet retreat and the subsequent exposure of its underlying shales have received attention to explain the CO$_2$ increases[10].

[1]Alfred-Wegener-Institut Helmholtz-Zentrum für Polar-und Meeresforschung (AWI), Bremerhaven 27568, Germany. [2]MARUM–Center for Marine Environmental Sciences and Faculty of Geosciences, University of Bremen, Bremen 28359, Germany. [3]Frontiers Science Center for Deep Ocean Multispheres and Earth System, and Key Laboratory of Marine Chemistry Theory and Technology, Ocean University of China, Qingdao 266100, China. [4]State Key Laboratory of Marine Geology, Tongji University, Shanghai 200092, China. [5]Korea Polar Research Institute, Incheon 21990, Republic of Korea. [6]Present address: Department of Environmental Science, Stockholm University, Stockholm 11418, Sweden. ✉e-mail: Junjie.Wu@aces.su.se; gesine.mollenhauer@awi.de; ru_st@uni-bremen.de

Oxidized petrogenic organic carbon ($OC_{petro}$; rock-derived OC that is typically radiocarbon free) can be reintroduced into the active carbon cycle through chemical weathering or microbial utilization[11,12]. Such processes have been proposed to regulate atmospheric $CO_2$ levels over geological timescales[13]. With more studies into this field, modern observations describe $OC_{petro}$ oxidative weathering as a supply-limited process, oxidation flux depending on the rock erosion rate[14,15]. High $OC_{petro}$ supply is found in glaciated regions today and glacial denudation of bedrock is a significant contributor. In the southern Alps of New Zealand, the $OC_{petro}$ oxidative weathering fluxes in glacier-dominated watersheds (up to 50 tC km$^{-2}$ yr$^{-1}$) are 2–3 times higher than those in less-glaciated watersheds[15]. Along the southeast Alaskan coast, the mass accumulation rates (MARs) of $OC_{petro}$ in glaciated fjords are significantly higher than those in non-glaciated fjords[16]. Notably, initial evidence of enhanced $OC_{petro}$ mobilization during the ice-sheet retreat has been seen in a sedimentary record obtained from the Bering Sea[17]. The above lines of evidence support rather substantial $OC_{petro}$ mobilization for the previously glaciated continents and imply extensive $OC_{petro}$ oxidation during the last deglaciation. It further raises questions, depending on the magnitude of carbon release, whether the processes are connected to the millennial-scale $CO_2$ increases during glacial terminations[10].

During the LGM, much of northern North America was covered by the Laurentide Ice Sheet (LIS) and Cordilleran Ice Sheet[18,19]. According to the underlying bedrock, northern North America can be subdivided into the $OC_{petro}$-rich western Canadian bedrock and the $OC_{petro}$-poor Canadian Shield. Ice-sheet retreat exposed the $OC_{petro}$-rich western Canadian bedrock during the last deglaciation[20], including shales, coal, and oil sands[10,21], and thus Blattmann[10] hypothesized that bedrock exhumation and the subsequent $OC_{petro}$ oxidation acted as a carbon source to the deglacial $CO_2$ rise. However, dedicated studies of this process are still restricted in numbers and regions, and more work is required to test this hypothesis. Studying marine sediment cores archiving land-derived OC from this region may provide further insight into the relationship between glacial retreat and $OC_{petro}$ remobilization during the last deglaciation.

Another terrestrial carbon pool that may be remobilized during the ice-sheet retreat is permafrost carbon, which has been freeze-locked for thousands of years and is vulnerable to degradation once thaws. Coastal erosion during the rapid sea-level rise, as a result of ice volume reduction, has been proposed to remobilize permafrost carbon and contribute to the deglacial $CO_2$ rise[17,22–24]. However, so far, most studies on permafrost carbon remobilization have been carried out along the shelves or slopes of the Laptev Sea, the East Siberian Sea, the Chukchi Sea, the Bering Sea, the North Pacific, and the Okhotsk Sea[17,23–27]. With the possible exception of the Bering Sea, the continental areas in the hinterland of these regions remained largely unglaciated during the LGM. Thus, a complex combination of processes during the last deglaciation, including water runoff, shelf flooding, and permafrost thawing in the interior, makes it difficult to unambiguously determine the process of coastal erosion. In contrast to the unglaciated regions, ice sheets in glaciated regions hindered vegetation development and the accumulation of carbon-rich permafrost deposits in the interior. This unique feature precludes signals from hinterland permafrost thawing and thus makes it an ideal region to study the process of coastal erosion.

Our study area in the Canadian Beaufort Sea is ideally located to elucidate both $OC_{petro}$ mobilization and the process of coastal erosion during the last deglaciation. In this work, we analyze biomarker contents and radiocarbon signatures of sedimentary OC and of specific terrigenous biomarkers in core ARA04C/37 from the Canadian Beaufort Sea (Fig. 1), spanning the last 14 kyrs[28]. Another set of samples from

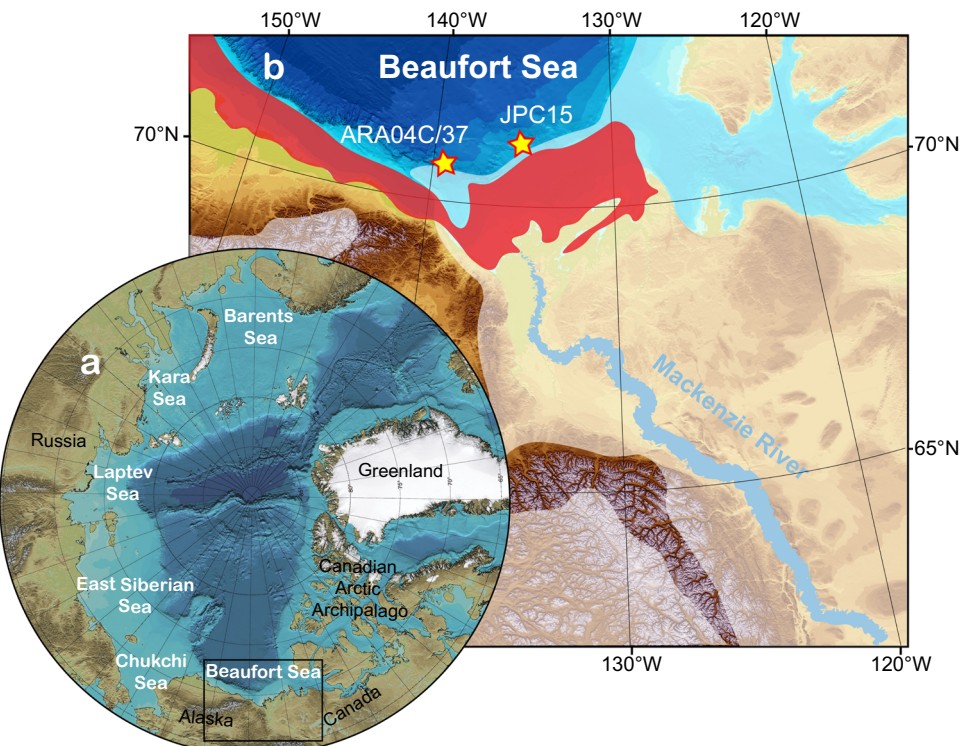

**Fig. 1 | Overview of the Arctic Ocean and core locations in the Canadian Beaufort Sea. a** overview of the Arctic Ocean and the study area (black box). **b** core locations of ARA04C/37 and JPC15 in this study are indicated by yellow stars and the Mackenzie River is outlined by a blue line. The red area between the paleo coast and the present coastline has been flooded since the Last Glacial Maximum (ca. 21 cal. kyr BP), according to the global ICE-6G_C model[49]. The white transparent areas indicate the Laurentide Ice Sheet extent during the Last Glacial Maximum[19]. The bathymetry is adapted from IBCAO Version 3.0[69].

nearby core JPC15[29] (Fig. 1) is analyzed to extend records into the Bølling/Allerød (B/A) interval. As ancient OC may consist of pre-aged terrestrial biospheric organic carbon ($OC_{terr-bio}$) and radiocarbon-free $OC_{petro}$, pyrolysis and biomarker proxies are used as supporting evidence of $OC_{petro}$ contributions. We demonstrate that ice-sheet retreat has caused substantial $OC_{petro}$ mobilization during the last deglaciation. Additionally, rapid sea-level rise at ca. 14 and 11 cal. kyr BP most likely caused two events of strong coastal erosion. Based on our findings, we attempt to estimate carbon release from $OC_{petro}$ oxidation and assess the impacts of oxidized $OC_{petro}$ and flooded permafrost on atmospheric carbon pools by using the global carbon cycle model BICYCLE[30]. The simulations suggest that the two processes additively may have a long-term effect of an increase in atmospheric $CO_2$ of 12 ppm and a decrease in atmospheric $\Delta^{14}C$ of 12 permil.

## Results

### Glacial retreat mobilized substantial amounts of petrogenic OC

Today, the Mackenzie River is the largest river in the Canadian Arctic, with a water discharge of $316 \text{ km}^3 \text{ yr}^{-1}$ and sediment flux of $124–128 \text{ Mt yr}^{-1}$ (ref. 31). Strongly influenced by Mackenzie River input, sedimentary OC in this region is predominantly terrigenous. In core ARA04C/37, hydrogen index and oxygen index derived from Rock-Eval pyrolysis indicate a dominant terrestrial OC source throughout the records (Supplementary Fig. 1), which is in agreement with the low values of $\delta^{13}C_{org}$ between −28.5‰ and −25.5‰ (Fig. 2d)[28].

Radiocarbon dating of bulk OC is used to characterize carbon age at the time of deposition (pre-depositional age). The pre-depositional ages of bulk OC were oldest (20–30 kyrs) between 14.5–10 cal. kyr BP while values decreased throughout the Holocene (Fig. 2c). We note that some samples from the late B/A interval show a low $^{14}C$ content close to the background. The pre-depositional ages in these samples are reported as minimum ages (>30 kyrs). The OC delivered into the Canadian Beaufort Sea was much older between 14.5 and 10 cal. kyr BP compared to the Holocene input. When compared to the mean age of modern Mackenzie River particulate organic carbon ($6.6 \pm 1.2$ kyrs) (ref. 32 and references therein), the differences are even larger.

Organic carbon in the modern Mackenzie River basin contains a large fraction of $OC_{petro}$[33,34], and the increased carbon age may suggest an even increased $OC_{petro}$ input in the past. Multiple proxies, i.e., carbon preference index of high molecular weight $n$-alkanes ($CPI_{alk}$), the fractional abundance of "biological" homohopane to its diagenetic isomers (fββ), and temperature at which pyrolysis yields the maximum of hydrocarbons ($T_{max}$), are used as support for large $OC_{petro}$ input between 14.5–10 cal. kyr BP. These proxies have been proposed to indicate carbon thermal maturity (see methods) and thus are indicative of $OC_{petro}$ input. However, as these proxies have limitations to specifically indicate $OC_{petro}$ input, interpretation of their variations warrants caution. Nevertheless, the overall low values of $CPI_{alk}$ and fββ, as well as high $T_{max}$ values, are consistent with $OC_{petro}$ input typical for the Mackenzie River basin and support increased $OC_{petro}$ contribution between 14.5–10 cal. kyr BP (Fig. 2e–g). Hence, the largely increased carbon age is most likely attributed to the enhanced $OC_{petro}$ contribution during the last deglaciation.

The period of enhanced $OC_{petro}$ contribution (14.5–10 cal. kyr BP) coincides with finely laminated sediments (Fig. 2c, h, Supplementary Fig. 2), implying higher $OC_{petro}$ MARs. The increased $OC_{petro}$ MARs likely resulted from enhanced $OC_{petro}$ supply. Glacial erosion and isostatic uplift due to the glacial retreat may have increased physical erosion of sedimentary rocks, resulting in a strongly enhanced $OC_{petro}$ supply. The ice sheets, which entrained abraded material from the underlying rocks, including ancient kerogen deposits during the glaciation, may have released $OC_{petro}$ upon melting. Besides, the possibly increased burial efficiency may have led to increased $OC_{petro}$ MARs. In this study, burial efficiency is defined as the ratio between OC release

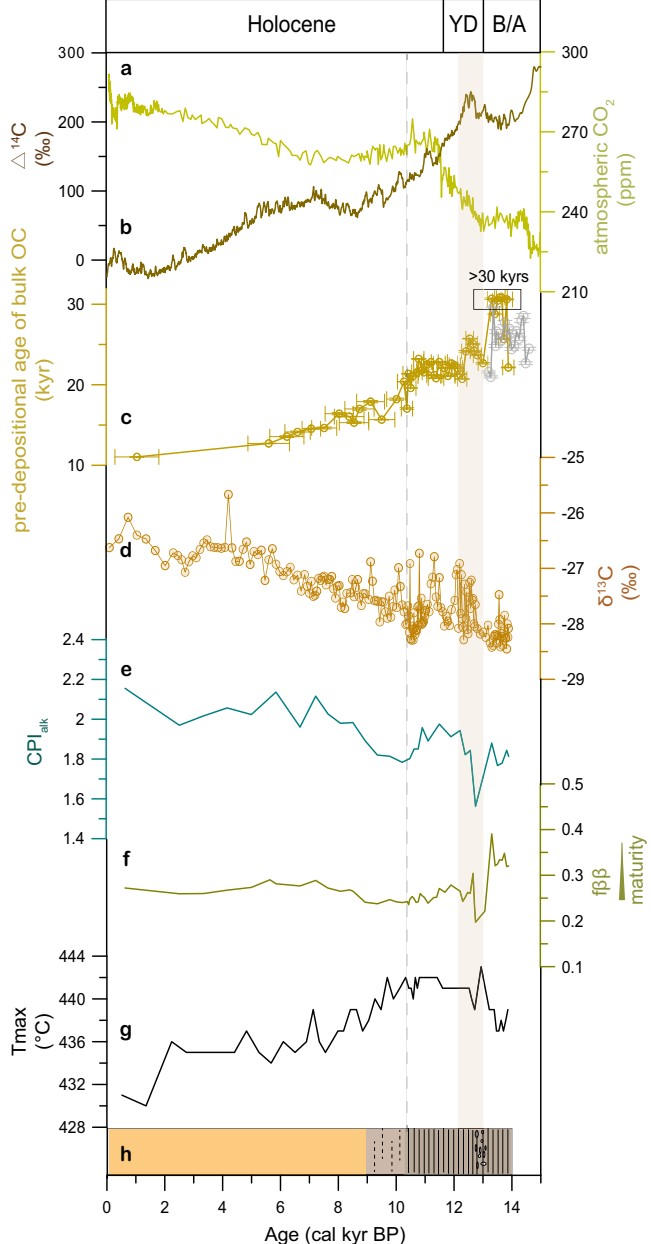

**Fig. 2 | Characteristics and thermal maturity of sedimentary organic matter in the Canadian Beaufort Sea. a** atmospheric $CO_2$ concentrations[70]; **b** radiocarbon content reconstructed in IntCal13[67]; **c** pre-depositional age of bulk OC, brownish yellow for core ARA04C/37 and gray for core JPC15. Error bars represent standard deviation; **d** $\delta^{13}C_{org}$ values in core ARA04C/37[28]; **e–g** thermal maturity proxies $CPI_{alk}$ (carbon preference index of high molecular weight n-alkanes), fββ (fractional abundance of "biological" homohopane to its diagenetic isomers), and $T_{max}$ (temperature at which pyrolysis yields the maximum of hydrocarbons) of core ARA04C/37; **h** lithology and sedimentary texture (according to Supplementary Fig. 2), dark brown section indicates finely laminated sediments and orange section indicate bioturbated silty clay. The light brownish section with dashed lines represents the transition characterized by weakly laminated sediments. Abbreviations in the figure are defined as YD Younger Dryas, B/A Bølling/Allerød.

on land and its long-term burial in marine sediments. The factors that may have increased burial efficiency are discussed below (see Discussion).

### Coastal erosion mobilized ancient permafrost carbon

The total organic carbon (TOC) MAR shows three distinct peaks during the last deglaciation and the early Holocene (Fig. 3d)[28].

The increased TOC MARs at the onset of the Younger Dryas (YD) can be explained by flood mobilization/transport of OC since multiple sedimentary records have documented the high-energy YD flood draining through the Mackenzie River to the Arctic Ocean (Fig. 3)[28,29]. However, the other two events may have different causes. Although the ice-margin retreat in the Fort McMurry area and evidence of gravels and an erosion surface in the Canadian Arctic Coastal Plain together suggest a post-YD flood originating from the proglacial lakes McMurray, Meadow, and Churchill between 11.7–9.3 cal. kyr BP[35,36], so far, an unambiguous meltwater flood signal has not been identified in marine records. For instance, core JPC15 only documented the YD freshening ($^{18}$O-depleted water), while $\delta^{18}O$ values during the putative post-YD flood are even higher than 2‰ (Fig. 3a)[29], the baseline of the B/A interval which represents a deglacial environment of meltwater discharge. In core ARA04C/37, the terrestrial biomarkers branched glycerol dialkyl glycerol tetraethers (brGDGTs, most probably derived from proglacial lakes) and the diol proxy ($F_{C32\ 1,15}$, indicative for running water) peaked during the YD flood, whereas no such clear signals were found during the post-YD flood (Fig. 3c, d), suggesting at least a less strong post-YD flood[28]. Therefore, the increase in TOC MAR at ca. 11 cal. kyr BP cannot be fully explained by a post-YD flood. To better constrain the carbon sources which contributed to high TOC MARs at ca. 14 and 11 cal. kyr BP, more information about terrestrial carbon is needed.

The high molecular weight fatty acids (HMW-FAs) are synthesized by terrestrial higher plants and are widely found in plants and soils. Since they are expected to be largely absent in mature petrogenic materials, HMW-FAs are commonly taken to represent $OC_{terr-bio}$[37–41]. The pre-depositional ages of HMW-FAs were the oldest (17–23 kyrs, with one age >30 kyrs) between 14.5 and 10 cal. kyr BP and values decreased during the Holocene (Fig. 3e). Furthermore, the ages of HMW-FAs during the last deglaciation exhibit large differences with modern $OC_{terr-bio}$ (5.8 ± 0.8 kyrs) in the Mackenzie River basin[34]. Because the remobilized $OC_{terr-bio}$ is always a mixture of constituents of different carbon ages, e.g., young biospheric organic carbon synthesized by plants and pre-aged biospheric carbon from relic permafrost or deeper soil profiles, the older mean age of HMW-FAs likely indicates lower contributions from young constituents. This can be attributed to the fact that the LIS has restricted vegetation and reduced contributions from freshly produced biospheric carbon during the last deglaciation. It implies that the mean age of $OC_{terr-bio}$ during the last deglaciation is different from the modern one.

The pre-depositional ages of HMW-FAs were not uniformly old between 14.5 and 10 cal. kyr BP. Ages were much younger during the YD, indicating a younger inland $OC_{terr-bio}$ transported by freshwater discharge. However, the pre-depositional ages of HMW-FAs sharply increased at 14 and 11 cal. kyr BP (Fig. 3d, e). As permafrost formation along the Mackenzie River basin was restricted by the LIS and there was no obvious enhancement in ice melting, the very old HMW-FAs and rapidly increased TOC MARs were unlikely caused by hinterland permafrost thawing or glacial melting. Rather, the two events co-occurred with the global meltwater pulses (MWP) 1a and 1b (Fig. 3d–f), indicating that the old HMW-FAs have a most probable source from coastal permafrost and/or ancient OC-rich deposits, released by coastal erosion. The first erosion event occurred slightly after the MWP 1a at ca. 13.9–13.4 cal. kyr BP (Figs. 3e, f). Since the nearby core JPC15/27 with a total recovery of 13 m has documented very high sedimentation rates and laminated sediments between 14.4–13.5 cal. kyr BP (6–12 m)[29], it is likely that the slight apparent delay in our records is due to a lack of accelerator mass spectrometry (AMS) $^{14}$C dates at the base of core ARA04C/37 (see methods) or an incomplete record of this event.

Although the distinct increases in TOC MAR were likely linked to coastal erosion during the MWP 1a and 1b, parameters of bulk OC, i.e.,

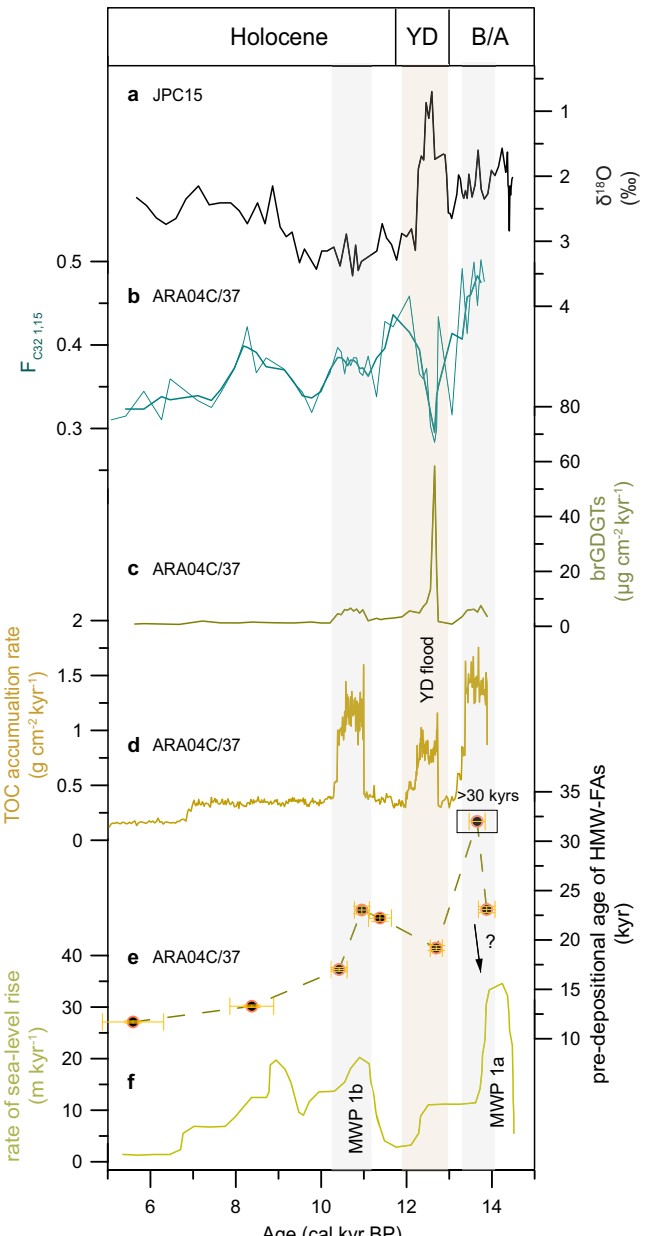

**Fig. 3 | Coastal erosion and Younger Dryas flood event. a** $\delta^{18}O$ values of *Neogloboquadrina pachyderma* in core JPC15[29]; **b, c** $F_{C32\ 1,15}$ (fractional abundance of $C_{32}$ 1,15-diol) and mass accumulation rate of brGDGTs (branched glycerol dialkyl glycerol tetraethers) in core ARA04C/37[28]; **d** mass accumulation rate of TOC (total organic carbon)[28]; **e** pre-depositional age of HMW-FAs (high molecular weight fatty acids). Horizontal error bars represent standard deviation, and vertical error bars represent the propagated uncertainties of the blank-corrected values; **f** rate of sea-level rise[71]. Abbreviations in the figure are defined as YD Younger Dryas, B/A Bølling/Allerød, MWP meltwater pulse.

$\delta^{13}C$, CPI, f$\beta\beta$, and $T_{max}$ did not show significant changes (Fig. 2). This suggests that, despite the strong coastal erosion, bulk OC was still dominated by $OC_{petro}$ while remobilization of ancient $OC_{terr-bio}$ only accounted for small contributions. The overwhelming $OC_{petro}$ input can be attributed to (1) continuously high $OC_{petro}$ supply from the hinterland and (2) significant contributions from the eroded coastal deposits containing $OC_{petro}$ debris[42].

## Discussion

The records of core ARA04C/37 present direct evidence for substantial remobilization of $OC_{petro}$ following the LIS retreat.

Towards quantitative estimates, a three-endmember mixing model based on a Markov chain Monte Carlo Bayesian approach was used to apportion relative contributions of $OC_{petro}$, $OC_{terr-bio}$, and marine biospheric carbon (Fig. 4b). More information on endmembers, endmember values, and the mixing model can be found in the Supplementary Discussion. The $OC_{petro}$ was estimated to contribute ~50–80% of TOC during the last deglaciation, and its relative contribution decreased to ~20–50% in the late Holocene (Fig. 4a, Supplementary Table 1). The estimated fraction in the late Holocene is close to the modern observations at the Mackenzie River delta, where $OC_{petro}$ accounts for ~10–30% of total particulate organic carbon[34]. In general, the $OC_{petro}$ fractions during the last deglaciation were one- to threefold higher than those in the late Holocene (Supplementary Table 1). In a conservative estimate, the TOC MARs during the deglaciation (~0.38 g cm$^{-2}$ kyr$^{-1}$) were ~six times higher than those in the late Holocene (~0.06 g cm$^{-2}$ kyr$^{-1}$) (see more information in Supplementary Discussion). We thus conclude that the $OC_{petro}$ MARs were 6–18 (or 12 ± 6) times higher during the last deglaciation.

The climate impact of $OC_{petro}$ exhumation during the last deglaciation is currently uncertain and depends on the fate of remobilized carbon. Through a simple pathway, the remobilized $OC_{petro}$ escaped remineralization and translocated from sedimentary rocks to oceanic storage which has negligible impacts on atmospheric $CO_2$ levels, while when through an oxidation pathway the remobilized $OC_{petro}$ was reintroduced into the carbon cycle which may affect atmospheric carbon chemistry. Emerging evidence couples $OC_{petro}$ oxidation flux with rock erosion rate[13], and particularly, the recent study observed higher $CO_2$ emissions in glaciated areas than in unglaciated areas[15]. The evidence provides us with the basis to explore an alternative scenario of large $OC_{petro}$ oxidation during the last deglaciation. Reconstructing oxidation of exhumed $OC_{petro}$ for past time periods is not possible from direct observations, but insights might be gained by studying sedimentary records. When postulating that our core record is representative of the process of petrogenic carbon mobilization in response to the deglaciation of North America, the record provides us with the opportunity to explore the effect enhanced rock erosion might have had on atmospheric $CO_2$.

Modern observations suggest that ~10–90% of the exhumed $OC_{petro}$ are oxidized, percentages varying in different regions, whereas $OC_{petro}$ buried in river/marine sediments is unoxidized and may be chemically and physically resilient[13]. Regarding the increased $OC_{petro}$ MARs as a consequence of enhanced rock erosion and assuming a constant burial efficiency, the $OC_{petro}$ oxidized should always keep pace with that accumulated in marine sediments. Therefore, the increase in $OC_{petro}$ MARs indicates an increase in $OC_{petro}$ oxidation flux by 6–18 (12 ± 6) times during the last deglaciation when compared to the modern oxidation flux. In the Mackenzie River basin, the modern $OC_{petro}$ oxidation fluxes are estimated to be ~0.45 ± 0.19 tC km$^{-2}$ yr$^{-1}$ for the Mackenzie River and a mean value of 0.89 ± 0.32 tC km$^{-2}$ yr$^{-1}$ for its main tributaries (0.94$^{+0.41}$/$_{-0.26}$, 0.78$^{+0.35}$/$_{-0.21}$, and 1.01$^{+0.42}$/$_{-0.25}$ tC km$^{-2}$ yr$^{-1}$ for the Peel, Arctic Red, and Liard River, respectively)[43]. The oxidation flux along the Mackenzie River is moderate in comparison to its tributaries. This can be explained by the fact that only 54% of the entire basin is on shales while all the tributaries drain through the shales-dominated mountain regions[44]. Therefore, the mean value (0.89 ± 0.32 tC km$^{-2}$ yr$^{-1}$) from the main tributaries is taken to represent the $OC_{petro}$ oxidation flux in shales regions, which is estimated to be 11 ± 7 tC km$^{-2}$ yr$^{-1}$ during the last deglaciation.

One should keep in mind that physical erosion was not the only factor responsible for the increases in $OC_{petro}$ MAR. Organic carbon burial efficiency increases with higher sedimentation rates[45,46]. On a larger spatial scale, global TOC MARs during the LGM were 147 ± 18% relative to the Holocene and then gradually decreased to the Holocene level[47]. This may indicate more efficient OC transfer from the sea surface to the seafloor and better preservation in marine sediment during the last deglaciation. In the study area, the meltwater-induced high discharge and shorter pathways from glaciers to the ocean may have further shortened the transport time, reducing OC oxidation during transport, and thus increased the OC burial efficiency in marine sediments. Besides, lower sea-level and closer distance of the study site to the coast may have caused higher $OC_{petro}$ MARs during the last deglaciation. While in opposite to the extensive shelves in the Laptev Sea or the East Siberian Sea, the offshore distance may change less in the Beaufort Sea due to the narrow shelf. Because the $OC_{petro}$ burial

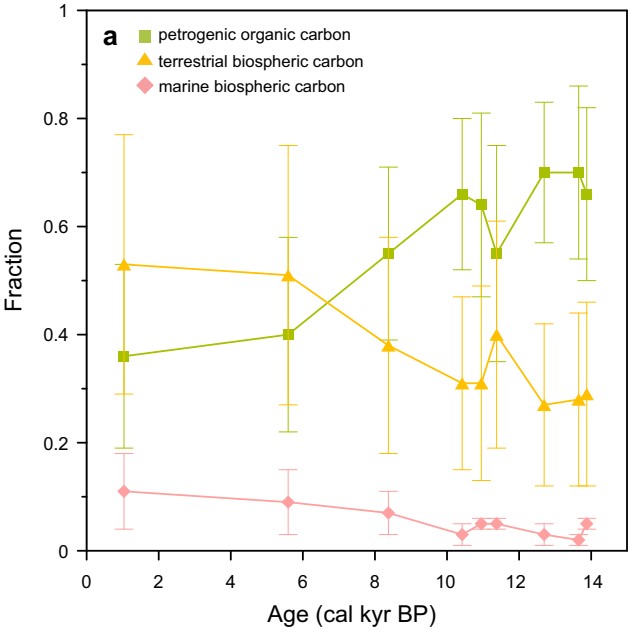
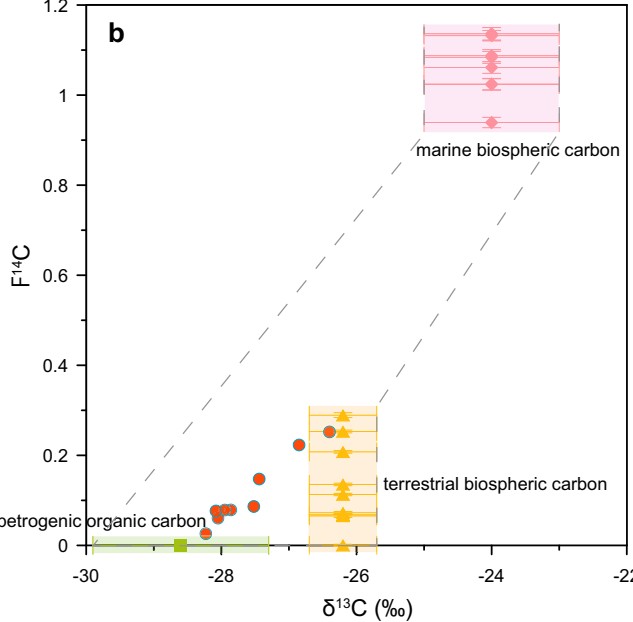

**Fig. 4 | Dual-isotope mixing model and source apportionments. a** OC fractions from different carbon sources. Error bars represent standard deviation; **b** δ¹³C and F¹⁴C values of measured samples (red dots) and of endmembers for marine biospheric carbon (pink diamonds), terrestrial biospheric carbon (yellow triangles), and petrogenic organic carbon (green squares) used in a dual-isotope mixing model. Error bars represent standard deviation.

efficiency might have increased via multiple processes during the last deglaciation, the estimated $OC_{petro}$ oxidation flux must therefore be regarded as an upper limit.

On the other hand, there are reasons that we would argue to expect even larger oxidation fluxes than we predicted. Glacial processes appear to enhance $OC_{petro}$ oxidation flux even at the same rate of physical erosion[15]. Besides, not all of the exhumed $OC_{petro}$ was transported to the ocean, and much of the glacially eroded material may have been deposited on land (e.g., moraines) or trapped in an intermediate pool of glacial lakes[48]. The enhanced $OC_{petro}$ MARs in marine records thus only reflect part of the exhumed $OC_{petro}$. These characteristics in the glacial watersheds imply that extrapolating oxidation flux from an unglaciated modern environment to a glaciated paleo scenario may cause an underestimate.

Outcropping shales that were exposed by glacial retreat after long-term isolation may have released additional $CO_2$ into the atmosphere. Estimating $OC_{petro}$ oxidation flux in such regions allows for a quantitative estimate of carbon release. The carbon yield ($J_{carbon}$) from $OC_{petro}$ oxidation can be estimated based on the exposed area ($A_{exposure}$), oxidative weathering fluxes ($F_{oxidation}$), and the exposure time ($T_{exposure}$) using the equation:

$$J_{carbon} = A_{exposure} \times F_{oxidation} \times T_{exposure} \qquad (1)$$

The $A_{exposure}$ is defined as freshly exposed areas with outcropping shales during the ice-sheet retreat. The area changes through time with regard to ice-sheet retreat, we, therefore, calculate $A_{exposure}$ based on shales distribution and the changes in ice-sheet extent[44,49] in North America (Fig. 5a; 180°W–75°W, 45°N–90°N). Because we assume that freshly exposed shales have higher oxidation flux and the flux decreases over time (discussed below), the $A_{exposure}$ is calculated in steps of 500 years to indicate the freshly exposed shales area in different time periods. The $F_{oxidation}$ is defined as $OC_{petro}$ oxidation flux for shales regions. Here we take $F_{oxidation}$ from the main tributaries of the Mackenzie River and assume it is applicable to entire North America. In the Mackenzie River basin, $F_{oxidation}$ is known to be $0.89 \pm 0.32$ tC km$^{-2}$ yr$^{-1}$ in the contemporary system[43] and is estimated (based on the increased $OC_{petro}$ MARs that reflect strongly increased erosion) to be $11 \pm 7$ tC km$^{-2}$ yr$^{-1}$ during the last deglaciation. From the last deglaciation towards the present, the large decrease in $F_{oxidation}$ may reflect processes such as soil formation and vegetation development, which reduce $F_{oxidation}$ with increasing exposure time. To include such processes in the calculation, we assume that $F_{oxidation}$ decreases with increasing substrate age in a similar manner as in silicate weathering (i.e., $F_{oxidation} = F_0 \times t^{-0.71}$)[50,51]. $F_0$ is the oxidation flux of freshly exposed substrate (10-year-old), and $F_{oxidation}$ denotes the oxidation flux of the substrate at age $t$. Calculation of $F_0$ requires a given $F_{oxidation}$ with a known substrate age $t$. Since Horan et al.[43] have estimated the modern oxidation flux ($0.89 \pm 0.32$ tC km$^{-2}$ yr$^{-1}$) for the shales-dominated region and the region is known to have been exposed between 15 and 10 cal. kyr BP[19], we thus assume for $10^4$ years old substrates ($t$) an oxidation rate of 1 tC km$^{-2}$ yr$^{-1}$ ($F_{oxidation}$) and obtain for newly exposed shales a rate of 21 tC km$^{-2}$ yr$^{-1}$ ($F_0$). Uncertainty bands are added to this approach by assuming that the modern $F_{oxidation}$ (-1 tC km$^{-2}$ yr$^{-1}$) is alternatively valid for substrate ages of 5 kyrs or 15 kyrs (Fig. 5b). Although the long-term behavior of $F_{oxidation}$ is implemented based on the modern oxidation flux, the oxidation fluxes during the last deglaciation that we derived from the data obtained on our sediment core (11 ± 7 tC km$^{-2}$ yr$^{-1}$) fall well within the range and correspond to a substrate age of <100 to 2000 years (indicated in Fig. 5b), to some degree supporting our assumption. Due to the variable $F_{oxidation}$ over time, carbon release ($J_{carbon}$) is calculated in steps of 100 years for entire North America and has been integrated over all time steps. The annual carbon release for entire North America is shown in Fig. 5c, with a cumulative release of 84 ± 30 PgC.

Besides the exhumation of $OC_{petro}$ during the last deglaciation, the events of increased TOC MARs also provide evidence for coastal erosion of permafrost that occurred in well-defined pulses. Because inland permafrost formation was restricted in the Mackenzie River basin, the maxima in accumulation occurring in the PreBoreal (MWP1b) and at or around MWP 1a were most likely attributed to the erosion of non-glaciated coastal permafrost (e.g., derived from Yukon coast), OC-rich deposits from the eastern coast, and residual permafrost/soil deposited in coastal regions. Within dating uncertainties, these two events were broadly coeval with those found in other records across the North Pacific and the Arctic Ocean[17,23,24,26]. Therefore, our records confirm that coastal erosion induced by rapid sea-level rise was a major process of aged permafrost carbon release in (sub-)Arctic coasts and shelves.

Remobilized permafrost carbon has been proposed to be highly bioavailable. A recent study mimics coastal permafrost erosion by incubating permafrost with and without seawater for the duration of one Arctic open-water season[52]. The authors demonstrate that substantial amounts of OC are quickly re-mineralized in all incubations, and the $CO_2$ production is even higher when seawater is added, indicating potentially substantial $CO_2$ emissions from erosion of permafrost onshore and within the nearshore waters. Estimates from Herschel Basin, a shelf basin in the Beaufort Sea, suggest that only ~40% of the eroded permafrost carbon from adjacent Herschel Island is buried locally in the basin[32], where it may, however, be further degraded. Another study based on OC concentrations of thawed samples from eroded bluffs estimates that ~66% of the OC contained in Yedoma deposits is released as $CO_2$ in one thawing season, prior to reaching a water body downslope[53]. Accordingly, the eroded permafrost carbon may have contributed to rapid deglacial $CO_2$ rises.

The two processes of carbon release, i.e., exhumation of $OC_{petro}$ and destabilization of coastal permafrost deposits, are both related to ice-sheet retreat. Ice-sheet retreat directly led to erosion, exposure, and oxidation of $OC_{petro}$ which released, depending on assumed substrate age, 84 ± 30 Pg of carbon. At the same time, the reduction in ice volume resulted in sea-level rise leading to coastal erosion/shelf flooding of permafrost from the Arctic shelves, which is estimated to release 85 Pg of carbon[24]. We applied the most recent version of the global carbon cycle model BICYCLE[30] in order to assess the impacts of ice-sheet retreat via the combined processes mentioned above on atmospheric $CO_2$ and $\Delta^{14}C$. The carbon release rate from $OC_{petro}$ oxidation is shown in Fig. 5c. Since our sedimentary records confirm coastal erosion during the rapid sea-level rise as a major process to release permafrost carbon in pulses, the carbon release rate from coastal permafrost follows the simulation in Winterfeld et al.[24], i.e., the release of 10 kyr-old permafrost carbon is restricted to a time window of 200 years for three pulses: 170 TgC yr$^{-1}$ (or 34 PgC) at 11.5 and 14.6 cal. kyr BP each and 85 TgC yr$^{-1}$ (or 17 PgC) at 16.5 cal. kyr BP. All carbon is released as $CO_2$ and directly enters the atmosphere in the model.

The results show three simulated peaks which are solely caused by the pulses of permafrost carbon release. These events include a first $CO_2$ peak of 3 ppm at 16.5 cal. kyr BP and two more $CO_2$ peaks of ~6 ppm at 14.6 and 11.5 cal. kyr BP (Fig. 5d). These permafrost carbon release pulses lead in the model to a decrease in $\Delta^{14}C$ of ~5 permil at 16.5 cal. kyr BP and of ~7–8 permil at 14.6 cal. kyr BP and 11.5 cal. kyr BP (Fig. 5e). The contribution of $OC_{petro}$ from North American shales adds a long-term rise of 4 ± 1 ppm to atmospheric $CO_2$ and of −4 ± 1 permil to atmospheric $\Delta^{14}C$. The simulated long-term effects over the last 20 kyrs when combining both processes are an increase in atmospheric $CO_2$ by 8 ppm and a decrease in atmospheric $\Delta^{14}C$ by 9 permil (Figs. 5d, e), explaining 10% and 2%, respectively, of the reconstructed changes in both variables.

Our attempt at a quantitative estimate of $OC_{petro}$ release, combined with coastal permafrost carbon release, demonstrates positive

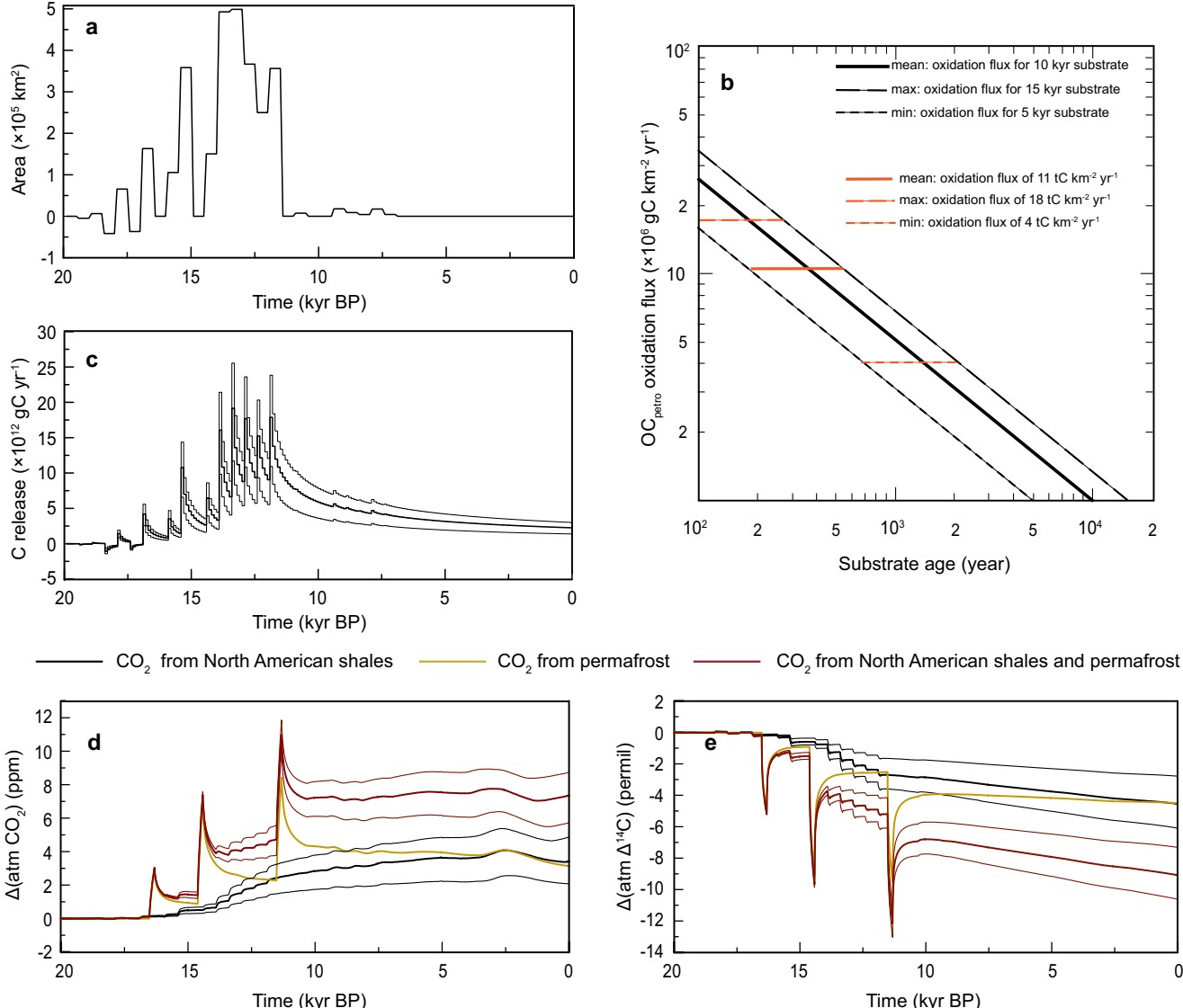

**Fig. 5 | Carbon release from OC$_{petro}$ oxidation and simulated impacts on atmospheric CO$_2$. a** areas of the ice-sheet retreat on shales calculated in steps of 500 years, based on the shales distribution and ice-sheet extent in North America[44,49]; **b** assumed long-term OC$_{petro}$ (petrogenic organic carbon) oxidation fluxes ($F_{oxidation}$) with substrate age ($t$) in North America, following the equation $F_{oxidation} = F_0 \times t^{-0.71}$. The black solid line demonstrates a scenario that the modern $F_{oxidation} = 1$ tC km$^{-2}$ yr$^{-1}$ is valid for shales of a substrate age of 10 kyrs, and the black dashed lines indicate scenarios that the modern $F_{oxidation} = 1$ tC km$^{-2}$ yr$^{-1}$ is valid for shales of a substrate age of 5 and 15 kyrs, respectively. Red lines mark the estimated past OC$_{petro}$ oxidation fluxes derived from the data of core ARA04C/37; **c** carbon release from OC$_{petro}$ oxidation, short-term peaks are attributed to the $J_{carbon}$ calculated in steps of 100 years for every 500-year exposed areas; simulated anomalies in **d** atmospheric CO$_2$ levels and in **e** atmospheric $\Delta^{14}$C using the global carbon cycle model BICYCLE-SE (ref. [30]) which also includes carbon release from thawing permafrost (ref. [24]) in three peaks (17|34|34 PgC).

feedback of the ice-sheet retreat that may partially explain the deglacial atmospheric CO$_2$ rise. Notably, OC$_{petro}$ release from North America is only one of the processes that are coupled with ice-sheet retreat, while the local net CO$_2$ budget must involve multiple other processes. Although sulfide weathering does not release CO$_2$, its product, sulfuric acid may increase carbonate solubility and thus boost CO$_2$ emissions from carbonate weathering, which is described as a supply-limited process[13,54]. In contrast, silicate weathering is known to be a process of long-term drawdown of atmospheric CO$_2$, and its associated carbon flux is sensitive to climate change and depends weakly on the erosion rate[13]. The burial of OC$_{terr-bio}$ by erosion is proven to be an important geological CO$_2$ sink[34]. By considering CO$_2$ emissions from oxidative weathering as well as CO$_2$ drawdown by OC$_{terr-bio}$ burial and silicate weathering, Horan et al.[43] have estimated for the modern Mackenzie River catchment a net transfer of CO$_2$ from the atmosphere to the

lithosphere, with a flux of -1 tC km$^{-2}$ yr$^{-1}$. Using modern unglaciated situations as analogs, the authors further postulate that the Mackenzie River catchment was a net CO$_2$ source during the LGM due to the reasons that (1) glacial erosion of sedimentary rocks may have strengthened oxidative weathering, (2) lower temperature may have weakened CO$_2$ drawdown by silicate weathering, and (3) limited vegetation and soils may have resulted in lower OC$_{terr-bio}$ stock in the landscape and weakened CO$_2$ drawdown by OC$_{terr-bio}$ burial. Our study presents direct evidence of substantial OC$_{petro}$ remobilization that may support the strengthened OC$_{petro}$ oxidative weathering during the last deglaciation. Besides, the very old OC$_{terr-bio}$ in our sediment archive most likely reflects limited vegetation in the glaciated environment (Fig. 3e), implying less fresh OC$_{terr-bio}$ burial. From a perspective of OC dynamics, our records support the Mackenzie River catchment as a net carbon source during the last deglaciation. However, the magnitude of

net $CO_2$ release from this region requires a comprehensive understanding of how ice-sheet retreat is involved in both inorganic and organic processes. Vegetation development and permafrost soil formation, in combination with weakening weathering processes, may have played a key role in transforming the region from a carbon source to a carbon sink during the glacial-interglacial transition.

Overall, the study provides evidence for enhanced $OC_{petro}$ exhumation and rapid coastal erosion events during the last deglaciation. Under the assumption of extensive $OC_{petro}$ oxidation during the glacial retreat, carbon release from the two processes contributes to the deglacial rise in atmospheric $CO_2$. The ice-sheet retreat has played a crucial role in mobilizing these aged/ancient terrestrial OC. These findings underscore the view that ice sheets may play an active role in the global carbon cycle during deglaciation. More work is needed to investigate carbon dynamics related to ice-sheet retreat, including oxidative weathering, silicate weathering, and biospheric carbon burial. Besides, it is essential to investigate carbon sources and carbon characteristics in other glaciated regions and to explore their roles in the deglacial carbon cycle.

## Methods

### Core location and sediment chronology
Gravity core ARA04C/37 was collected during the Araon Cruise ARA04C[55] at the Beaufort Sea continental slope, with a recovery of 595 cm (Fig. 1; 70°38.0212'N, 139°22.0749'W; 1173 m). The age-depth model of core ARA04C/37 has been established by Wu et al.[28] based on AMS$^{14}$C dating on calcareous foraminifera and, in the uppermost centimeters, excess $^{210}$Pb. This study updates the age-depth model by dating planktic foraminifera from the B/A interval, adding two new AMS$^{14}$C dates to the age-depth model (Supplementary Table 2). The base of core ARA04C/37 could not be dated due to a lack of planktic foraminifera. Therefore, the age model from Wu et al.[28], which is based on the correlation of magnetic susceptibility data between ARA04C/37 and nearby core JPC15 (Fig. 1), is adopted without further modification for the depths below 530 cm. Because our study focuses partly on the centennial rapid events, we used the Marine13 calibration curve to keep consistency with other published records[17,24].

Core JPC15 was obtained at the continental slope east of Mackenzie River (Fig. 1; 71°06.222'N, 135°08.129'W; 690 m). Samples from the B/A interval of core JPC15 were analyzed for this study. For more information and chronology of sediment core JPC15, please refer to Keigwin et al.[29].

### Rock-Eval pyrolysis
Rock-Eval pyrolysis was performed on bulk sediment samples according to Espitalie et al.[56]. Hydrogen and oxygen contents of the samples, measured as hydrocarbon-type compound and carbon dioxide yields respectively, were normalized to organic carbon and displayed as hydrogen index (mgHC/gC) and oxygen index (mgCO$_2$/gC). In a van Krevelen-type diagram, a classification illustrating carbon types is possible[57]. Furthermore, the temperature at which pyrolysis yields the maximum of hydrocarbons ($T_{max}$) is used as an indicator of the thermal maturity of the kerogen. Immature organic matter usually has $T_{max}$ values of <435 °C.

### Biomarker analyses and thermal maturity indicators
Freeze-dried sediments (~5 g) were extracted with DCM:MeOH (2:1, v/v), and an internal standard (Squalane, 2.4 ug/sample) was added prior to analytical treatment. Total lipid extracts were concentrated and separated into a hydrocarbon fraction (containing $n$-alkanes and hopanes) and an alcohol fraction via open column chromatography with silica gel (6 mm i.d.*4.5 cm). Hydrocarbon fractions were eluted with 5 ml $n$-hexane, followed by alcohol fraction elution with 9 ml ethyl acetate:$n$-hexane (1:4, v/v).

The $n$-alkanes were analyzed using a gas chromatograph (GC, Agilent 7890 A) coupled to a flame ionization detector (GC-FID). Homohopane isomers were analyzed with an Agilent 6850 gas chromatograph (GC) coupled to an Agilent 5975 C VL MSD quadrupole mass spectrometer operating in electron impact ionization (70 eV) and full-scan (m/z 50–600) mode.

The $n$-alkanes were identified with external standards, and the carbon preference index of $n$-alkanes ($CPI_{alk}$) was calculated as follows (Eq. 2):

$$CPI_{alk} = \frac{1}{2} \times \left( \frac{C23 + C25 + C27 + C29 + C31}{C22 + C24 + C26 + C28 + C30} + \frac{C23 + C25 + C27 + C29 + C31}{C24 + C26 + C28 + C30 + C32} \right) \tag{2}$$

$CPI_{alk} > 3$ is indicative for significant contributions of fresh OC from immature deposits, whereas $CPI_{alk}$ close to 1 is indicative for a dominance of thermally mature OC (ref. 17 and references therein). Because $CPI_{alk}$ may also vary with OC degradation state[58], a combination with the relative abundances of homohopane isomers (fββ) can further indicate contributions from thermally mature OC.

Homohopane isomers ($C_{31}$) were identified by relative retention times and mass spectra[17]. The relative abundance of the "biogenic isomer" 17β,21β(H) 22 R ($C_{31}$ββR) to the "diagenetic isomers" 17β,21α(H) 22 R ($C_{31}$βαR), 17β,21α(H) 22 S ($C_{31}$βαS), 17α,21β(H) 22 R ($C_{31}$αβR), and 17α,21β(H) 22 S ($C_{31}$αβS) is described as Eq. 3[17]:

$$fββ = \frac{C31ββR}{C31ββR + C31αβS + C31αβR + C31βαS + C31βαR} \tag{3}$$

Values of 1 indicate the absence of "diagenetic isomers", whereas values of 0 indicate the absence of "biogenic isomer".

### Radiocarbon analyses of bulk OC and HMW-FAs
Radiocarbon dating was performed following methods described in Mollenhauer et al.[59]. Briefly, according to the respective TOC contents, sediment samples containing ~1 mg OC were weighed into silver capsules and were acidified with 6 N hydrochloric acid (HCl) to completely remove the inorganic carbon. Acid evaporation was conducted on a hot plate at 60 °C, and the dried samples were then stored in an oven (60 °C) until the analysis. Samples including the silver capsules were packed into tin capsules and combusted individually via an Elementar vario ISOTOPE EA (Elemental Analyzer). Oxidized carbon ($CO_2$) was directly graphitized by the Ionplus AGE3 system (Automated Graphitization System)[60]. Radiocarbon contents of samples were analyzed using the Ionplus MICADAS dating system[61,62].

For compound-specific (HMW-FAs) radiocarbon dating, sediment samples (~50–70 g) were extracted with DCM:MeOH (9:1, v/v) using a Soxhlet for over 48 h. The total extracts were hydrolyzed with potassium hydroxide (KOH, 0.1 M) in MeOH:H$_2$O (9:1, v/v) at 80 °C for 2 h. The neutral lipids were extracted with $n$-hexane, and the $n$-alkanoic acids were then extracted with DCM after adjusting the pH to a value of around 2 by the addition of 37% HCl. The extracted $n$-alkanoic acids were methylated with 37% HCl and MeOH with a known $^{14}$C-signature in a nitrogen atmosphere at 80 °C for over 12 h. The fatty acid methyl esters (FAMEs) were extracted with $n$-hexane and subsequently separated from polar compounds by silica gel chromatography.

FAMEs with chain lengths >C$_{24}$ were purified by preparative capillary gas chromatography (PC-GC) using an Agilent 6890 N GC equipped with a Gerstel Cooled Injection System (GIS) and connected to a Gerstel preparative fraction collector[63]. The GC was equipped with a Restek Rxi-XLB fused silica capillary column (30 m, 0.53 mm i.d., 0.5-μm film thickness). All samples were injected stepwise with 5 μL per injection. Afterwards, the purified individual FAMEs were transferred into tin capsules and packed. Samples were combusted via the

Elementar vario ISOTOPE EA (Elemental Analyzer), and the isotopic ratios ($^{14}C/^{12}C$) of produced $CO_2$ were determined via the directly connected AMS, the MICADAS system, which is equipped with a gas-ion source.

Radiocarbon contents of the samples were analyzed along with reference standards (oxalic acid II; NIST 4990c) and blanks (phthalic anhydride; Sigma-Aldrich 320064) and in-house reference sediments. Blank correction and standard normalization were performed via the BATS software[64]. All results are reported as fraction modern carbon ($F^{14}C$).

## Blank assessment and corrections
Compound-specific samples analyzed for radiocarbon are sensitive to contamination during processing (procedure blank), for example, the carbon introduced via column bleed and carry-over as well as from the tin capsules. Blank correction of compound-specific samples requires careful determination of $F^{14}C$ of the blank and the size of blank. For this purpose, radiocarbon analyses of in-house reference samples from $^{14}C$-free Eocene Messel Shale ($F^{14}C_{OC} = 0$) and modern apple peel ($F^{14}C_{OC} = 1.029 \pm 0.001$) processed in the same way as the compound-specific samples were conducted to determine the $F^{14}C$ and mass of blank, following the method of Sun et al.[65]. All radiocarbon data were corrected for the procedural blank and, to remove the contribution of the methyl group added during the derivatization process, a methyl correction was further performed through isotopic mass balance. Uncertainties were fully propagated.

## Pre-depositional ages of OC
The pre-depositional age of OC can be derived from the initial radiocarbon content ($\Delta^{14}C_{initial}$; radiocarbon content prior to the OC deposition to marine sediment), which has been calculated based on the following equation:[66]

$$\Delta^{14}C_{initial} = (F^{14}C\, e^{\lambda t} - 1) \times 1000‰ \qquad (4)$$

$F^{14}C$ is the measured fraction modern carbon, and for compound-specific samples, the blank- and methanol-corrected $F^{14}C$ values are used. $\lambda$ is the decay constant of radiocarbon, and $t$ is the time since deposition (according to the core chronology).

The radiocarbon content of the past atmosphere differed from the modern atmosphere, and thus the atmospheric radiocarbon content at the time of deposition ($\Delta^{14}C_{atm}$) was taken from the atmospheric $\Delta^{14}C$ record of IntCal13 (for consistency, as Marine13 was used for the age model)[67]. The apparent conventional $^{14}C$ age (pre-depositional age) thus has been calculated by the following equation:[66]

$$\begin{aligned} pre-depositional\,age = &-8033 \times \ln\,[(1 + \Delta^{14}C_{initial}/1000)/ \\ &(1 + (\Delta^{14}C_{atm}/1000)] \end{aligned} \qquad (5)$$

## Carbon cycle modeling
We use the most recent version of the global carbon cycle model BICYCLE[30] to simulate the impact of the released carbon on atmospheric $CO_2$ and $\Delta^{14}C$. This version, called BICYLE-SE, includes solid Earth processes (volcanic $CO_2$ outgassing, continental weathering, shallow water carbon sink in coral reefs) and a process-based sediment model, which calculates as function of the depth-dependent carbonate ion concentrations the accumulation or dissolution of $CaCO_3$ in the deep ocean sediments. The core of the model still consists of a 10-box ocean, a 1-box atmosphere, and a 7-box terrestrial biosphere. We calculate the contribution from $OC_{petro}$ released from a retreating LIS on shales (Fig. 5c) and permafrost[24], by additional carbon fluxes to a control run, which is based on the scenario SE in Köhler and Munhoven[30] with the exception of a passive (constant) terrestrial carbon pool. This has been chosen to make simulation results comparable

to Winterfeld et al.[24], in which a similar setup, but using an older model version, has been applied. The $^{14}C$ production rate is kept constant at 25% higher than preindustrial levels in order to obtain at LGM an atmospheric $\Delta^{14}C$ of ~400 permil, comparable to reconstructions. Simulations are initialized with interglacial conditions, and the state variables of the sediment are taken after an 800 kyr-long spinup, and started at 210 cal. kyr BP to give the model enough time to run into a reasonable state which is largely independent from initial conditions and spinup.

While we use three different versions of the fluxes for $OC_{petro}$ depending on the assumed substrate age (5, 10, 15 kyrs), we use only one version of the permafrost-based carbon release related to coastal erosion. Here, 17, 34, and 34 Pg of 10 kyr-old carbon are released in three 200-year-long pulses starting at 16.5, 14.6, and 11.5 cal. kyr BP[24]. The carbon released from both processes is entering the model as ingoing $CO_2$ fluxes to the atmosphere. The time-dependent forcing—the changes in climatic boundary conditions (e.g., temperature, sea-level, sea ice, ocean circulation, iron input to the marine biology)—is otherwise identical to what is described in Köhler and Munhoven[30].

## Data availability
The data generated in this study have been deposited in the PANGAEA repository (https://doi.org/10.1594/PANGAEA.939847)[68]. Source data are provided with this paper.

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

## Acknowledgements

We gratefully thank the professional support of the captain and crew of RV Araon on the expedition ARA04C in 2013. We acknowledge Lloyd D. Keigwin for providing study material from core JPC15. We thank Walter Luttmer for conducting Rock-Eval pyrolysis. Thanks to Vera Meyer for input and laboratory support. Furthermore, we thank Torben Gentz, Elizabeth Bonk, and Maylin Malter for radiocarbon analyses. We acknowledge the Alfred Wegener Institute Helmholtz Center for Polar and Marine Research (AWI) and China Scholarship Council for financial support. This work contributes to PALMOD, the German Paleomodeling Research Project funded by BMBF. This research is also supported by the Basic Core Technology Development Program for the Oceans and the Polar Regions (Grant NRF-2021M1A5A1075512 to S.N.) from the National Research Foundation of Korea funded by the Ministry of Science and ICT (MSIT).

## Author contributions

R.S. and G.M. designed this study. S.N. performed field work and sampling and provided the X-Ray digital photographs. J.W. carried out biomarker analyses, pre-treatment of bulk OC, and identification of foraminifers. J.H. conducted purification and separation of HMW-FAs. G.M. and H.G. performed AMS[14]C analyses and [14]C data processing. R.S. provided and evaluated the Rock-Eval pyrolysis data. P.K. estimated the carbon release and performed carbon cycle modeling. J.W. wrote the manuscript with input from G.M., R.S., K.F., and B.W. All co-authors contributed to the manuscript at different stages.

## Funding

## Competing interests

The authors declare no competing interests.
