## [Peer Review File · Nature Communications]

Deglacial release of petrogenic and permafrost carbon from the Canadian Arctic impacting the carbon cycleReviewer #1 (Remarks to the Author):

Review of the manuscript Deglacial release of petrogenic and permafrost carbon from the Canadian Arctic impacting the carbon cycle by Wu et al., submitted as research paper to Nature Communications (NCOMMS-21-36247)

The paper presents a 14-kyr long sediment record from the Beaufort Sea continental margin that is used to reconstruct terrestrial organic carbon (terrOC) release from Arctic Canada during the last deglaciation. By analyzing a range of organic geochemical proxies, including bulk carbon isotopes ^{13}C and ^{14}C , as well as terrestrial biomarkers (high molecular weight (HMW) n-alkanes and n-alkanoic acids, hopanes, branched glycerol dialkyl glycerol tetraethers – brGDGTs), the authors describe sources and mass accumulation rates of organic carbon (OC) to the marine seabed. This receptor-based approach permits studies of OC release over a large catchment area over the course of the climate history of the past 14 kyrs, which includes parts of the Bølling/Allerød the early Holocene warming periods. The isotopic fingerprint of the bulk OC and (compound-specific) ^{14}C ages of terrestrial biomarkers suggest that large parts of the OC that was released during the last deglaciation originated from a strongly pre-aged terrestrial source, which points at petrogenic rocks and thawing permafrost as possible sources of the terrOC. Based on back-of-the-envelope calculations and a carbon cycle model, the authors show that large-scale terrOC oxidation and CO_2 release could have contributed to the deglacial rise in atmospheric CO_2 .

The study addresses an important knowledge gap in climate change research: To constraint changes to the carbon cycle as a result of a warming Arctic climate. This study adds to an increasing number of studies from the North Pacific (Winterfeld et al., 2018; Meyer et al., 2019) and the Arctic Ocean (Tesi et al., 2016; Keskitalo et al., 2017; Martens et al., 2019, 2020), which combine to provide evidence for large-scale permafrost thawing and terrOC release in the Arctic during past warming events. As a novelty, this record is the first to present deglacial terrOC release for the Canadian Arctic, which seems to be characterized by different terrOC sources and remobilizing dynamics when compared with the Eurasian Arctic. The present study thereby provides important insight on how contemporary and future Arctic warming, including further deglaciation and sea-level rise, may affect large-scale carbon cycling and contribute to climate-carbon feedback e.g. from permafrost thawing. The methods involved in this study are sound and I'm convinced that the data produced is of high quality. However, the paper contains a number of conceptual and interpretative weaknesses, for which I recommend major revisions before this work can be considered for publication in Nature Communications. These major and more minor comments are described below.

Overarching points

- 1. The authors hypothesize that oxidation of petrogenic OC released large amounts of CO_2 to the atmosphere and thereby contributed to the rise in CO_2 during the last deglaciation. This builds on the finding that mass accumulation rates of petrogenic OC to Beaufort shelf sediments were >10 times higher during deglacial warming periods than today, which suggests much higher petrogenic OC release from land at that time. Further, the authors assume that higher petrogenic OC release was accompanied by significant CO_2 release. This assumption follows a hypothesis that is currently under debate, e.g. in the non-peer-reviewed preprint in ref. 13, which states that petrogenic OC oxidation may have contributed to the deglacial rise in CO_2 . However, this concept is highly uncertain. Indeed, it seems that there is scattered evidence that glacial excavation of bedrock in glaciated areas may release petrogenic OC and locally cause higher CO_2 emissions than in non-glaciated areas (e.g., Horan et al., 2017). Yet, the exposure of fresh bedrock and possibly petrogenic OC is outbalanced by the drawdown of atmospheric CO_2 through silicate weathering, a process that is known to contribute to long-term removal of atmospheric CO_2 and modulating the Earth climate system over**

glacial-interglacial time scales. In addition to chemical weathering, changes to in the biosphere and inland permafrost may have profound impact on the OC balance throughout the last deglaciation. This is even specifically stated for the Mackenzie drainage by the literature cited in the paper (Horan et al., 2019; ref 43 in the manuscript). Unfortunately, these processes are not well considered or appropriately discussed in the present version of the paper. Furthermore, the budget calculations for the cumulative release of 84 Pg as CO₂ during the last deglaciation is insufficiently described. Given the lack of observational evidence and in the light of the published literature on this topic, I'm skeptical about the author's findings about significant release of CO₂ from petrogenic OC oxidation.

2. The authors leverage dual-isotope source apportionment (¹³C and ¹⁴C) to distinguish between past release of terrOC from petrogenic sources and permafrost. As end member for petrogenic OC, the mixing model includes published ¹³C ratios of source rocks within the Mackenzie river catchment and assumes ¹⁴C-free OC. As end member of the terrestrial biosphere, ¹³C ratios of soils and pre-depositional ¹⁴C ages of HMW n-alkanoic acids are used. There are a number of issues with these end members I would like to address:

i) The chosen end members do not account for terrOC release from older permafrost deposits from the non-glaciated parts or OC-rich glacial till in Arctic Canada, while such deposits are present in this area, e.g. on Hershel Island and along the Yukon coast, and contribute to significant terrOC release.

ii) These end members cannot distinguish between any terrOC from soils or other permafrost deposits that were buried below the Laurentide Ice Sheet. Would this be another possible source of terrOC that may have contributed CO₂ during the last deglaciation?

Further, the compound-specific ¹⁴C age as end member for the terrestrial biosphere is problematic for several reasons:

iii) HMW n-alkanoic acids may not solely represent "fresh" biospheric terrOC but is most likely a mixture of terrOC integrated from several sources over this large area. To my knowledge, it may also not be ruled out that these compounds partly derive from petrogenic OC directly, or are affected and contaminated by microbial recycling of petrogenic OC.

iv) HMW n-alkanoic acids represent only a small fraction of the terrestrial organic matter but is here used to represent terrOC at the bulk level. HMW n-alkanoic acids are, however, comparatively degradation-resistant and may survive longer in the environment than others, which may create a bias towards higher ¹⁴C ages. For the revised version of this paper, the authors should consider an improved end member selection for the source apportionment of the OC.

Minor points

Line 25: As stated above, I see only limited support for this statement.

Line 54-57: The studies mentioned (ref. 10 and 16) provide only evidence for petrogenic OC release. I see no support for "substantial OCpetro mobilization and oxidation...".

Line 66-67 "more work is required to test this hypothesis": That's also my understanding.

Introduction: The introduction needs to explain how CO₂ is produced from petrogenic OC. Is it only a process of chemical weathering or is also microbial decomposition involved?

Line 135: How is burial efficiency defined in this study? Does it represent i) the preservation of OC between release on land and long-term burial in marine sediments or ii) the loss of OC between sediment deposition at the sediment-water interface and residual OC buried during early diagenesis processes?

Line 155: Consider a different word choice than catastrophic here and elsewhere in the paper.

Line 161: How old were the oldest HMW-FA? This needs to be stated to follow the meaning of this sentence.

Line 167: How is the composition expected to be different that it would result in a different end member age?

Line 176-178: The "delay" of an event may also result from lateral transport of sediment OC. Could this process affect the dating of this or other events in this study?

Line 179 "Although the distinct increases in TOC MAR were linked...": add "likely" or similar. I suggest to use more careful phrasing here and elsewhere in the paper.

Line 180: What parameters of bulk OC? Looking at the $\delta^{14}C$ of bulk OC I see very large changes (i.e. a drop of the pre-depositional age from ~ 30 to ~ 20 kyr). How do you explain this difference if it's not a change in source or composition?

Line 213 and 219-222: Wouldn't also the CO₂ drawdown by chemical weathering increase? As described above, I think this needs to be included in this discussion to reflect on the net CO₂ exchange with the atmosphere.

Line 234-236: We can or we may expect? This and other sentences of this paragraph are rather speculative and need to be more carefully phrased.

Line 236: Sediments reactive surface areas are also considered to sequester OC to form mineral-organic associations, which stabilize and protect OC from remineralization. I would argue that this process rather limits the reactivity of the OC.

Line 243-252: What area (km²) was used here? Are glaciated areas considered to produce CO₂ as well? Was the area allowed to change through time with regard to glacier retreat? More background information is needed to follow this calculation.

Line 256: What is the flux estimate for 10-yr old shales based on? This number is 20 times higher than the measured/estimated CO₂ flux in this region. If this is only assumed I think an alternative scenario needs to be presented, in which the age of the substrate has no effect to the oxidation rate/CO₂ production. This may also replace the uncertainty calculations in line 257-259.

Line 262: Is this 84 Pg budget based on entire North America or only the Mackenzie river catchment? This does not become clear in this paragraph or elsewhere in the manuscript.

Line 304: What are these figures of released permafrost OC based on and why are they divided as such portions between the different warming periods? Is this based on the inundated shelf area and permafrost OC release by coastal erosion in the Arctic Ocean? It appears that most of the Arctic Ocean shelf area was flooded after 11.7 kyr BP (e.g. for the Laptev Sea: Bauch et al., 2001).

References

Bauch, H., Mueller-Lupp, T., Taldenkova, E., Spielhagen, R., Kassens, H., Grootes, P., Thiede, J., Heinemeier, J., Petryashov, V., 2001. Chronology of the Holocene

transgression at the North Siberian margin. *Glob. Planet. Change* 31, 125–139. [https://doi.org/10.1016/S0921-8181\(01\)00116-3](https://doi.org/10.1016/S0921-8181(01)00116-3)

Horan, K., Hilton, R.G., Dellinger, M., Tipper, E., Galy, V., Calmels, D., Selby, D., Gaillardet, J., Ottley, C.J., Parsons, D.R., Burton, K.W., 2019. Carbon dioxide emissions by rock organic carbon oxidation and the net geochemical carbon budget of the Mackenzie River Basin. *Am. J. Sci.* 319, 473–499. <https://doi.org/10.2475/06.2019.02>

Horan, K., Hilton, R.G., Selby, D., Ottley, C.J., Gröcke, D.R., Hicks, M., Burton, K.W., 2017. Mountain glaciation drives rapid oxidation of rock-bound organic carbon. *Sci. Adv.* 3. <https://doi.org/10.1126/sciadv.1701107>

Keskitalo, K., Tesi, T., Bröder, L., Andersson, A., Pearce, C., Sköld, M., Semiletov, I.P., Dudarev, O. V., Gustafsson, Ö., 2017. Sources and characteristics of terrestrial carbon in Holocene-scale sediments of the East Siberian Sea. *Clim. Past* 13, 1213–1226. <https://doi.org/10.5194/cp-13-1213-2017>

Martens, J., Wild, B., Muschitiello, F., O'Regan, M., Jakobsson, M., Semiletov, I., Dudarev, O. V., Gustafsson, Ö., 2020. Remobilization of dormant carbon from Siberian-Arctic permafrost during three past warming events. *Sci. Adv.* 6, 6546–6562. <https://doi.org/10.1126/sciadv.abb6546>

Martens, J., Wild, B., Pearce, C., Tesi, T., Andersson, A., Bröder, L., O'Regan, M., Jakobsson, M., Sköld, M., Gemery, L., Cronin, T.M., Semiletov, I., Dudarev, O. V., Gustafsson, Ö., 2019. Remobilization of Old Permafrost Carbon to Chukchi Sea Sediments During the End of the Last Deglaciation. *Global Biogeochem. Cycles* 33, 2–14. <https://doi.org/10.1029/2018GB005969>

Meyer, V.D., Hefter, J., Köhler, P., Tiedemann, R., Gersonde, R., Wacker, L., Mollenhauer, G., 2019. Permafrost-carbon mobilization in Beringia caused by deglacial meltwater runoff, sea-level rise and warming. *Environ. Res. Lett.* 14, 085003. <https://doi.org/10.1088/1748-9326/ab2653>

Tesi, T., Muschitiello, F., Smittenberg, R.H., Jakobsson, M., Vonk, J.E., Hill, P., Andersson, A., Kirchner, N., Noormets, R., Dudarev, O. V., Semiletov, I.P., Gustafsson, Ö., 2016. Massive remobilization of permafrost carbon during post-glacial warming. *Nat. Commun.* 7, 13653. <https://doi.org/10.1038/ncomms13653>

Winterfeld, M., Mollenhauer, G., Dummann, W., Köhler, P., Lembke-Jene, L., Meyer, V.D., Hefter, J., McIntyre, C., Wacker, L., Kokfelt, U., Tiedemann, R., 2018. Deglacial mobilization of pre-aged terrestrial carbon from degrading permafrost. *Nat. Commun.* 9, 3666. <https://doi.org/10.1038/s41467-018-06080-w>

Reviewer #2 (Remarks to the Author):

Wu et al. investigated two cores collected from the Beaufort Sea to study the carbon dynamics since the last deglacial time. They concluded that there is a significant amount of CO₂ released through the oxidation of petrogenic carbon and the erosion of permafrost. They tried to build on the previous theory framework and made quantitative calculations to demonstrate the importance of these two processes in explaining the variation of atmospheric CO₂ concentrations and C isotopes. Overall, the authors presented attractive and robust data echoing those published previously, and this study further confirms the theory of terrestrial OC oxidation contributing to deglacial atmospheric CO₂ fluctuations. Here, I am providing some concerns, questions, and suggestions that may help improve the MS.

Major questions:

1) Petrogenic carbon's oxidation and permafrost OC's erosion have previously been proposed as possible mechanisms substituting the ocean ventilating theory in explaining

deglacial atmospheric CO₂ fluctuations. This study tried to make sense quantitatively; while reading through the MS, I doubted the calculation, relatedness, and contribution from this study. First of all, the estimate of the amount of OCpetro bases heavily on an equation published in a Biogeoscience Discussion, which is still not accepted by Journal Biogeosciences. As a critical parameter of this equation, the oxidation rate (0.89) was adopted from another study (Horan et al., 2019). However, while reading through this key reference, the only place that mentioned this same oxidation rate was under the discussion of carbonate weathering. Additionally, the calculation of weathering rate is not constant, as deglacial retreat would largely thicken the soil and thus decrease the weathering rate. Overall, I doubt the calculated 84.3PgC release through the OCpetro erosion. Furthermore, I do not see the linkage between the budget calculation and the core measurements from this study. This study is on the sedimentary record from the Beaufort Sea, while the budget calculation of OCpetro oxidation is purely based on hinterland data published previously. Second, this study further reinforced the importance of permafrost erosion as an essential source of atmospheric CO₂ during the deglacial time. However, I do not see how much this study contributed rather than repeating what was published by Winterfeld et al. (2018). Overall, I doubt the carbon budget calculation and the contribution solely from this study.

2) It is always important to consider water depth variation while interpreting data generated from a sediment core offshore. Since the last glacial time, the sea level has increased by ~170m. During the deglacial time, the location where sediment cores were retrieved was much shallower. What matters more is that the sampling locations are closer to the coast. Such deglacial-interglacial differences could largely explain the sedimentation rate differences. Explicitly, the sedimentation rate is expected to be much higher when the sampling location was closer to the coastline during the deglacial time. It essentially explains higher OCpetro and OCpermafrost MARs at the deglacial time. Additionally, the results and discussion rely heavily on one sediment core, while the other core is only presented with limited data and discussion.

3) The end-member model and the associated results doubted me. First of all, the $\delta^{13}\text{C}$ was adopted from Hilton et al. (2015), who further cited Johnston et al. (2012). Johnston did report $\delta^{13}\text{C}$ values of sedimentary rocks in this region; unfortunately, all the $\delta^{13}\text{C}$ numbers are from the Neoproterozoic strata, which do not represent the average sedimentary rock $\delta^{13}\text{C}$ values in this region with the widespread presence of Phanerozoic strata. Second, $\Delta^{14}\text{C}$ values vary for terrestrial end-members over time; therefore, the authors picked $\Delta^{14}\text{CFA}$ as $\Delta^{14}\text{C}$ values for soil end-member values. It is commonly perceived that there is an offset between bulk and CSRA $\Delta^{14}\text{C}$ values, and such offsets vary with time, especially in this region. Additionally, all data numbers closely cluster together (Fig. 4b); hence, the error generated from the three end-member-model could be enormous.

4) The application of some "classic" parameters. Rock-Eval is commonly applied as a method in estimating the thermal maturity of source rocks. However, Rock-Eval analysis on modern biomass, soil, and sediments generates similar T_{max} values. The observed variation of T_{max} downcore could fully be explained by the diagenetic alteration over the past 14kyr and is not a robust proxy in demonstrating petrogenic carbon inputs. Besides, $f_{\beta\beta}$ is not valid when you assume the petrogenic carbon is mature or over-mature already since thermal cracking breaks down organic molecules. Additionally, the soil has been found to produce significant amount of $\beta\alpha$ hopenes, rather than $\beta\beta$ hopenes, even when the soil is "fresh".

Other comments:

Line 57: The above evidence only supports mobilization, but not oxidation of OCpetro.

Lines 87-89: thermal maturity based on Rock-Eval, in this study, does not indicate petrogenic OC inputs.

Lines 204: deglacial does not cause physical erosion; glacier advancing does.

Lines 204-209: shorter transport distance is more reasonable than the intense physical erosion of bedrock in explaining enhanced OCpetro MAR.

Lines 207-209: The burial efficiency might be higher during the deglacial time since these particles have been deposited along floodplains and have gone through preliminary oxidation.

Lines 219-240: this section presented a contradictory discussion on whether OCpetro is

extensively protected or weathered. The first part argues that OCpetro is protected due to mineral protection, while the second part claims that the OCpetro is heavily weathered due to high surface area.

Lines 239-258: The discussion of the budget is based on several criteria, which essentially is not related to the core of this study, which is what generated from the sediment cores.

Lines 241: Blattmann (2021) was not accepted and should not serve as an essential reference here. Additionally, I do not see this equation being presented in Blattmann (2021, biogeoscience discussion).

Line 251: This is not the number from the original study (Horan et al., 2019).

Lines 259-273: the discussion of permafrost relies heavily on Winterfeld et al. (2018); hence it is hard to extract the new contribution from this study.

Line 268-270: residual permafrost/soil on the floodplain or what has been deposited at the coastal region could supply aged permafrost carbon to the study area. This permafrost has been exposed to the atmosphere or has gone through fluvial transport and hence is more resistant to degradation.

Reviewer #3 (Remarks to the Author):

Reviewer #3 attachment on the following page.

Dear Wu et al.,

The manuscript presents a record of organic matter burial near the mouth of the Mackenzie River covering the episode of most recent deglaciation towards the present day. The authors characterize the organic matter using radiocarbon and biomarkers. From this, a timeline of changing sources and burial rates of permafrost, OC_{petro} , marine, and terrestrial biospheric OC is reconstructed. From the burial rates, OC_{petro} oxidation rates in the catchment are estimated and together with published data a quantitative estimate on the combined contribution of permafrost and OC_{petro} derived CO_2 is generated which is placed into a global context.

The work presents a novel dataset from a remote location that is strategically chosen to provide best possible insight into the dynamics of deglaciation. The authors have chosen to interpret their dataset in a completely new way with respect to the dynamic behavior of OC_{petro} both in terms of its reburial flux and its decay along its source-to-sink trajectory. These are basic constraints that are conspicuously missing and just beginning to emerge in the literature (e.g., Berg et al., 2021). For the first time, the study of Wu et al. provides us with a view of changes taking place across an entire cycle of deglaciation in an area recording the history of the largest continental ice sheet retreat on Earth.

In my opinion, the current manuscript is in need of major structural improvement and refinement of arguments and technical precision. By addressing the comments elaborated below, the current work can greatly increase its persuasiveness and potential to energize the community to investigate the dynamic role of terrestrial stocks of permafrost and petrogenic OC on atmospheric chemistry and climate across glacial-interglacial cycles. This work is of great value for providing new perspectives and basic quantitative constraints on the carbon fluxes that have governed the fate of our planet in our geologically most recent history.

Sincerely,

Thomas Blattmann

23.11.2021 Zurich

Major comments and suggestions:

The Results section and Discussion section should either be strictly separated or merged. Currently, considerable discussion resides in the results part (e.g., 132-141, and many other bits and pieces) and leads to some repeats in the manuscript. In conjunction with this, the permafrost vs. OC_{petro} components of the interpretation are presented in a confusing manner and I got the impression that the authors were unsure about their own interpretation. Regarding the presentation of the results, if the authors decide to merge their results and discussion, I suggest that the authors use the bulk radiocarbon data to build their core story and use other datasets (e.g., dated compounds, GDGTs, etc.) as complementary components to build the discussion. The bulk radiocarbon data seem to provide a clear interpretation around which the complementary datasets provide interesting discussion avenues.

The definition of terrestrial biosphere and distinguishing or equating that with permafrost carbon is unclear. For example, in lines 173-174 the HMW-FAs were interpreted as stemming from permafrost and immature OC_{petro} . However, in the source apportionment in lines 452-454, HMW-FAs ages are used to

represent the age of terrestrial biospheric organic carbon. So is terrestrial biospheric organic carbon equated with permafrost carbon? And in this case, is OC_{petro} also considered terrestrial biospheric carbon of ancient origin? Overall, the usage of the terms “ancient”, “terrestrial biospheric organic carbon”, “permafrost”, etc. lead to some confusion. I recommend the authors refine the usage of their terminology to make the interpretation clearer.

How was the Monte Carlo simulation implemented? How was the uncertainty of the marine reservoir age considered? Seems like two ages were considered, but how was this implemented numerically? Was the uncertainty of the age model of the core also incorporated in the calculation of uncertainty for the different pools of OC? Given the uncertainty of the age model at the base of the core(s), this seems warranted.

What is the uncertainty/reproducibility of the Rock-Eval results for T_{max} ? For a non-expert, the approximately 12°C spread in T_{max} (Fig. 2g) seems narrow and a discussion of the uncertainty appears warranted.

Please include (representative) chromatograms for fatty acids, alkanes, etc. for documentation purposes in the supplementary information. Additionally, I felt like the visual display of total mass accumulation rates and total organic carbon content was missing for interested readers to get a feel for the data. I think these data would reside well in the supplementary information. Furthermore, I imagine that a graphical display of mass accumulation rate of the different types of TOC with the resolution made possible by the source partitioning presented in Fig. 4 would also be a valuable addition to the supplemental and could be helpful in aiding the discussion.

The quantitative estimates of carbon release from permafrost appears to solely come an Asian-based study (Winterfeld et al., 2018) and OC_{petro} oxidation presented largely hinges on assuming a constant burial efficiency and assuming that the sedimentary records are representative for the integrated catchment (sediment dispersal or sorting seems unaddressed). The restricted occurrence of permafrost in the Mackenzie Basin is mentioned (e.g., lines 171, 273), so is the permafrost-released CO_2 from this area considered negligible? Additionally, OC_{petro} mass accumulation rates from the site investigated is claimed to represent deglaciated North America (lines 310-311), which misses inputs from the Alaskan fjords and Canadian west coast (Cui et al., 2016) and from the catchments draining into the Hudson Bay, which also drain OC_{petro} -rich areas of the Western Canadian Sedimentary Basin, as well as moraines (line 241) and glacial lakes (Blattmann et al., 2019) which trap OC_{petro} in these intermediate sinks. Skeptical readers may have a hard time with the patchwork carbon budget and buying into the idea of the constant burial efficiency, but I think there is an opportunity to present this as a conservative estimate.

Minor comments:

The terms remobilization and oxidation are sometimes used imprecisely. For example, in line 27 remobilization is used. However, remobilization per se is a carbon neutral process and in its context a positive climate feedback is proposed. There are a few such instances in the manuscript. Please refine the usage of these terms.

There are several points of discussion surrounding the mode of permafrost mobilization/decay, including coastal erosion, shelf flooding, and hinterland thawing. However, from a carbon budget perspective, it is

unclear why distinguishing between these different modes of permafrost mobilization and decay is important. I think adding reasons or motivation behind answering these questions would be valuable for the readers.

Line 36: Anomaly of the atmosphere or ocean? Please specify.

Line 135: In this context, how relevant is isostatic uplift over this timescale?

Line 147: What kind of fieldwork evidence is this referring to? Please specify.

Line 149: Delete “yet”. This word implies an expectation that may or may not agree with reality.

Line 151: What sort of baseline? Recorded in what? Please clarify or edit.

Line 230, 231: Comment: OC_{petro} in many cases probably already comes associated with lithogenic minerals (e.g., mica and chlorite, Blattmann et al., 2019, Blattmann & Liu 2021)

Line 262: I suggest reducing the number of “significant” digits to 84 ± 30 PgC.

Lines 298-304: This sentence needs to be fixed.

Line 330: I suggest deleting “Unfortunately”.

Line 452: Concerning the notation: I suggest using separate notation for age-corrected ^{14}C concentrations, as I assume you are using age-corrected ^{14}C values for assessing the end members?

Lines 131, 473: Reference 32 seems like a weak as this study looked at a specific time in the Neoproterozoic. Large swaths of the catchment are Cretaceous and Devonian (see geological map of Canada). Recovering data hidden in other studies seems worthwhile to get a representative picture on the stable carbon isotope composition of OC_{petro} , or discussing the representativity of the available constraints.

Language comments:

The singular and plural forms for bedrock are the same: bedrock. See dictionaries Merriam Webster, etc.

Please refine text for smoother reading. For example:

Line 63: delete “within the deglaciated terrains”

Line 65: delete “might have” (hypothesis implies this)

Lines 71, 72: move “during the last deglaciation” to the end of the sentence

Please review the usage of articles (the, a, an) and possibly have a native speaker proofread.

References

- Berg, S., Jivcov, S., Kusch, S., Kuhn, G., White, D., Bohrmann, G., Melles, M. and Rethemeyer, J. (2021) Increased petrogenic and biospheric organic carbon burial in sub-Antarctic fjord sediments in response to recent glacier retreat. *Limnology and Oceanography*.
- Blattmann, T.M., Wessels, M., McIntyre, C.P. and Eglinton, T.I. (2019) Petrogenic organic carbon retention in terrestrial basins: A case study from perialpine Lake Constance. *Chemical Geology* 503, 52-60.
- Blattmann, T.M., Liu, Z., Zhang, Y., Zhao, Y., Haghypour, N., Montluçon, D.B., Plötze, M. and Eglinton, T.I. (2019) Mineralogical control on the fate of continentally derived organic matter in the ocean. *Science* 366, 742-745.
- Blattmann, T.M. and Liu, Z. (2021) Proposing a classic clay mineral proxy for quantifying kerogen reburial in the geologic past. *Applied Clay Science* 211, 106190.
- Cui, X., Bianchi, T.S., Jaeger, J.M. and Smith, R.W. (2016) Biospheric and petrogenic organic carbon flux along southeast Alaska. *Earth and Planetary Science Letters* 452, 238-246.

Author responses to reviewer comments on the manuscript entitled “*Deglacial release of petrogenic and permafrost carbon from the Canadian Arctic impacting the carbon cycle*” (NCOMMS-21-36247) by J. Wu and co-authors

We thank all three reviewers for the efforts spent on reviewing and we greatly appreciate the constructive and helpful comments. All comments were carefully considered and most of them were incorporated into the revised version of the manuscript. We believe the current version has been significantly improved based on these comments.

We made substantial revisions according to the reviewers’ major concerns, which are:

1. All reviewers have major concerns about our mixing model, endmembers, and endmember values. We thus compiled a Supplementary Text to discuss in detail how we define endmembers and endmember values (updated) and how we test the robustness of mixing model outcomes. The outcome is also updated in the Main Text.
2. All reviewers raise concerns about our budget calculation and have questions about the linkage between the budget calculation and the measurements made on the sediment samples. We think this is because of an insufficient description of our approach. Therefore, we made major revisions to more clearly explain the concept of our approach.
3. Reviewer#1 points out weaknesses in our argument, particularly considering the scattered evidence of higher OC_{petro} oxidation in glaciated regions. In the revised version, we re-structured our argument and emphasize that our intention is to explore the potential magnitude of OC_{petro} release based on the emerging evidence rather than derive hard numbers. Besides, a key part missing in the initial version is the discussion of other weathering processes as well as $OC_{\text{terr-bio}}$ burial. We thus added discussions of these processes to the Main Text.
4. Reviewer#2 points out the limitations of some classic parameters to specifically indicate thermal maturity. We carefully changed our interpretation of their variations and used them more as supporting evidence.
5. We added graphical displays in the Supplementary Information, e.g., chromatograms, total MARs, and the MARs of different types of carbon.

Below we present our point-by-point response to the reviewers’ comments.

Reply to Reviewer #1:

1.1 “The authors hypothesize that oxidation of petrogenic OC released large amounts of CO₂ to the atmosphere and thereby contributed to the rise in CO₂ during the last deglaciation. This builds on the finding that mass accumulation rates of petrogenic OC to Beaufort shelf sediments were >10 times higher during deglacial warming periods than today, which suggests much higher petrogenic OC release from land at that time. Further, the authors assume that higher petrogenic OC release was accompanied by significant CO₂ release. This assumption follows a hypothesis that is currently under debate, e.g. in the non-peer-reviewed preprint in ref. 13, which states that petrogenic OC oxidation may have contributed to the deglacial rise in CO₂. However, this concept is highly uncertain. Indeed, it seems that there is scattered evidence that glacial excavation of bedrock in glaciated areas may release petrogenic OC and locally cause higher CO₂ emissions than in non-glaciated areas (e.g., Horan et al., 2017). Yet, the exposure of fresh bedrock and possibly petrogenic OC is outbalanced by the drawdown of atmospheric CO₂ through silicate weathering, a process that is known to contribute to long-term removal of atmospheric CO₂ and modulating the Earth climate system over glacial-interglacial time scales. In addition to chemical weathering, changes to in the biosphere and inland permafrost may have profound impact on the OC balance throughout the last deglaciation. This is even specifically stated for the Mackenzie drainage by the literature cited in the paper (Horan et al., 2019; ref 43 in the manuscript). Unfortunately, these processes are not well considered or appropriately discussed in the present version of the paper. Furthermore, the budget calculations for the cumulative release of 84 Pg as CO₂ during the last deglaciation is insufficiently described. Given the lack of observational evidence and in the light of the published literature on this topic, I’m sceptical about the author’s findings about significant release of CO₂ from petrogenic OC oxidation.

Reply 1.1: We thank the reviewer for these insightful comments, pointing out the rather limited database on the process of OC_{petro} oxidation following glacial excavation, and highlighting the counter-acting process of silicate weathering. We would like to stress, though, that we do by no means ignore the effects of silicate weathering, but that we attempt an estimate of the potential impact the commonly not considered process of petrogenic OC oxidation might have.

During the LGM, the bedrock was overlain by the Laurentide Ice Sheet and Cordilleran Ice Sheet which were up to a thousand-meter thick. The complete retreat of ice sheets must have eroded and exposed the long-term isolated OC_{petro} to the atmosphere. We observed significant OC_{petro} mobilization, while whether it was largely oxidized is unclear due to the currently incomplete understanding of this process. Unfortunately, we are not able to study such extreme processes in the contemporary Earth system or in the next few centuries. Therefore, here we demonstrate our attempt to explore an alternative scenario of large CO₂ emissions in response to enhanced erosion, based on evidence from past periods of rapid warming and glacial melting. We argue that the OC_{petro} flux ($10.7 \pm 6.6 \text{ tC km}^{-2} \text{ yr}^{-1}$) we estimated for the last deglaciation is within a reasonable range compared with the compilation of Blattmann (2022)¹ according to which OC_{petro} oxidation fluxes from rock-disseminated forms of kerogen under aerobic conditions amount to 0.3-64 MgC/km²/yr and the fluxes from catchments with ongoing deglaciation range from 9-50 MgC/km²/yr². In the Main Text, we re-structured our argument

and clarified that our study makes an attempt at exploring a new scenario. Please see lines 212-223.

We agree that multiple weathering processes and $OC_{\text{terr-bio}}$ burial must have been involved and contributed to the net CO_2 budget. Our study, based on organic carbon dynamics, focuses on the climate impact of the single process of OC_{petro} oxidation, while the discussion of the net carbon budget is missing in our initial version. We added more discussions on these processes and the net CO_2 budget during the glacial period. Our study may contribute to the discussions on OC-related processes, while a comprehensive understanding of the role of glacial retreat or a net carbon budget needs more studies on both inorganic and organic processes. Please see our discussion in lines 343-370.

Besides, we would like to point out that the reference of Blattmann (2022) is now published as a peer-reviewed paper: Blattmann, T. M.: Ideas and perspectives: Emerging contours of a dynamic exogenous kerogen cycle, *Biogeosciences*, 19, 359–373, <https://doi.org/10.5194/bg-19-359-2022>, 2022.

1.2 The authors leverage dual-isotope source apportionment (^{13}C and ^{14}C) to distinguish between past release of terrOC from petrogenic sources and permafrost. As end member for petrogenic OC, the mixing model includes published ^{13}C ratios of source rocks within the Mackenzie river catchment and assumes ^{14}C -free OC. As end member of the terrestrial biosphere, ^{13}C ratios of soils and pre-depositional ^{14}C ages of HMW n-alkanoic acids are used. There are a number of issues with these end members I would like to address:

Reply: As Reviewers #1 and #2 have major concerns about our mixing model, we here explain our ideas before replying to each comment. In general, there are two ways to define endmembers in a mixing model. One is based on the type of terrOC, i.e., petrogenic or biospheric (e.g., ref^{3,4}). Notably, in such a definition, terrOC is apportioned regardless of the place of origin. Another approach is based on the spatial distribution of terrOC. For example, Grotheer et al. (2020)⁵ define terrOC endmembers in the Beaufort region by attributing carbon to the Mackenzie River catchment or Herschel Island. However, in such a definition, a complex mix of organic matter including petrogenic OC is characterized in each system and we cannot estimate the relative contributions of different types of organic material independently. Both ways of definitions can be used for a mixing model but both have limitations. Defining endmembers based on their spatial distributions usually cannot cover all possible sources, whereas when defining endmembers by their origins it is usually difficult to distinguish OC_{petro} and old $OC_{\text{biosphere}}$.

There are two key reasons why we believe it is better here to define endmembers according to their origin. First, the key finding of this study is the enhanced petrogenic OC input during the last deglaciation, therefore our aim is to estimate its contributions. Second, the molecular-level radiocarbon dating ($F^{14}CFA$) can represent all possible $OC_{\text{terr-bio}}$ sources as a whole and help to distinguish between OC_{petro} and old $OC_{\text{terr-bio}}$.

i) The chosen end members do not account for terrOC release from older permafrost deposits from the non-glaciated parts or OC-rich glacial till in Arctic Canada, while such deposits are

present in this area, e.g. on Hershel Island and along the Yukon coast, and contribute to significant terrOC release.

Reply 1.2-i: We respectfully disagree with this comment and would like to elaborate on our approach. Our assumption is based on the fact that functionalized compounds like fatty acids are considered to be absent in mature organic materials like petrogenic carbon that has undergone diagenesis and partly even katagenesis. On the other hand, long-chain n-alkyl compounds are widely used biomarkers for terrestrial higher plants, and their compound-specific age can reflect combined contributions to sediment from a variety of thermally immature sources including vegetation, soils, permafrost, or even older deposits like peats or lignite⁶⁻⁸. Therefore, the compound-specific radiocarbon ages we obtained on long-chain fatty acids are taken to reflect the mean ¹⁴C content of the complex mixture of all materials rich in terrestrial organic matter contributing to our sediments, including old permafrost deposits or OC-rich glacial till.

Further, we suspect that Reviewer#1 suggests us to include ¹³C values from Herschel Island and Yukon coast when we define the endmember values for terrestrial biospheric carbon. In the revised version, we include ¹³C values from the Herschel Island retrogressive thaw slumps (n=7) and onshore samples from the Yukon Coastal Plain (n=19)⁹. Overall, the mean ¹³C value is still within the range defined in our initial version. Please see Supplementary Information for more details.

ii) These end members cannot distinguish between any terrOC from soils or other permafrost deposits that were buried below the Laurentide Ice Sheet. Would this be another possible source of terrOC that may have contributed CO₂ during the last deglaciation?

Reply 1.2-i: As elaborated above, our study aims at distinguishing different types of organic matter and not different regional contributions. The hypothetical OC-rich permafrost deposits buried below the Laurentide Ice Sheet could be a possible carbon source once the glacial retreat has exposed it to the atmosphere, and we acknowledge that we cannot distinguish its contribution from permafrost that did not reside below a glacier.

We did not evaluate the significance of such hypothetical subglacial organic-rich permafrost relicts due to the following reasons. First, in most of the glaciated regions, organic-rich permafrost deposits did not form, and only the regions that remained unglaciated during any of the glacial stages might form such organic-rich permafrost. According to the ice-sheet configuration described in Batchelor et al. (2019)¹⁰, such areas in North America are much smaller than in other regions and mainly existed in the south and east of North America. Our core location is not suitable to document such changes. Considering that such organic-rich permafrost deposits are typically unconsolidated, we estimate their potential to be covered by glaciers rather than being eroded completely during the glacial advance to be rather unlikely. Second, our records show relatively small biospheric OC fractions during the last deglaciation (even during the rapid coastal erosion events), suggesting an overall smaller contribution of permafrost carbon at least in Northern North America. Due to these reasons, our material might not be ideal to address the importance of permafrost carbon in this region.

Further, the compound-specific ^{14}C age as end member for the terrestrial biosphere is problematic for several reasons:

iii) HMW n-alkanoic acids may not solely represent “fresh” biospheric terrOC but is most likely a mixture of terrOC integrated from several sources over this large area. To my knowledge, it may also not be ruled out that these compounds partly derive from petrogenic OC directly, or are affected and contaminated by microbial recycling of petrogenic OC.

iv) HMW n-alkanoic acids represent only a small fraction of the terrestrial organic matter but is here used to represent terrOC at the bulk level. HMW n-alkanoic acids are, however, comparatively degradation-resistant and may survive longer in the environment than others, which may create a bias towards higher ^{14}C ages.

Reply 1.2-iii and 1.2-iv: We agree that there are shortcomings of HMW n-alkanoic acids to represent the terrestrial biospheric carbon. As mentioned in comments 1.2-iii and 1.2-iv, the HMW n-alkanoic acids may contain (small) petrogenic contributions and only represent a small fraction of terrOC which may be comparatively degradation-resistant. These shortcomings may cause a bias towards higher ^{14}C ages. Despite that, we here explain the reasons and advantages of using HMW n-alkanoic acids as endmembers as follows:

(1) In the study of Drenzek et al. (2007)⁴ (from the Beaufort Sea), it has been found that “In this case, however, the depleted isotopic compositions of the pyrolysis products suggest that long-chain fatty acids and alkanes are likely representatives of terrestrial OC on the whole”.

(2) Importantly, the vegetation, permafrost, and ice sheets in North America were dynamic during the last deglaciation. The relatively fixed endmember values from the contemporary system cannot reflect such changes and thus might not be applicable to the paleo system. The HMW n-alkanoic acids, as a whole to represent all $\text{OC}_{\text{terr-bio}}$ sources (e.g., freshly produced OC from plants, $\text{OC}_{\text{terr-bio}}$ from permafrost deposits, and $\text{OC}_{\text{terr-bio}}$ from variable soil profiles in non-permafrost regions), are expected to reflect the dynamic changes in terrestrial biospheric carbon composition over time, although we acknowledge that one single compound group does not represent the full diversity of organic matter present in such a system.

(3) As mentioned above, HMW n-alkanoic acids represent all possible sources as a whole, relieving the constraint to find proper endmember values to represent all possible sources.

(4) Endmember values determined on bulk OM on land are fixed while the radiocarbon signals of terrestrial OC may change (e.g., via degradation and re-suspension) during transport to the core location. This may cause an underestimate of the $\text{OC}_{\text{terr-bio}}$ contribution. In contrast, using HMW n-alkanoic acids values determined in the study material circumvents these possible additional complications.

Please see Supplementary Information for discussions on defining $\text{OC}_{\text{terr-bio}}$ endmember values.

For the revised version of this paper, the authors should consider an improved end member selection for the source apportionment of the OC.

In the Supplementary Information, we updated the definitions or added discussions on ^{13}C values for $\text{OC}_{\text{terr-bio}}$ and OC_{petro} (explained below) endmembers. As for ^{14}C values for $\text{OC}_{\text{terr-}}$

bio endmember, as mentioned above, we believe the signals in HMW n-alkanoic acids have obvious advantages over published bulk-level endmember values.

Minor comments

Line 25: As stated above, I see only limited support for this statement.

We agree with this comment and rephrased it. Please see lines 26-27.

Line 54-57: The studies mentioned (ref. 10 and 16) provide only evidence for petrogenic OC release. I see no support for “substantial OC_{petro} mobilization and oxidation...”.

We agree ref. 10 and 16 only provide evidence for OC_{petro} remobilization. Based on the study of Horan et al. (2017)² and the study of Hilton and West (2020)¹¹ which describe OC_{petro} oxidation as a supply-limited process, we may imply a higher CO₂ emission under such conditions. We rephrased it. Please see lines 65-66.

Line 66-67 “more work is required to test this hypothesis”: That’s also my understanding. Introduction: The introduction needs to explain how CO₂ is produced from petrogenic OC. Is it only a process of chemical weathering or is also microbial decomposition involved?

We thank for this comment and we add in the Introduction an explanation of the CO₂ release from OC_{petro}. Both chemical weathering and microbial decomposition may be involved. Please see lines 53-56.

Line 135: How is burial efficiency defined in this study? Does it represent i) the preservation of OC between release on land and long-term burial in marine sediments or ii) the loss of OC between sediment deposition at the sediment-water interface and residual OC buried during early diagenesis processes?

The “burial efficiency” in this study indicates the preservation of OC between release on land and long-term burial in marine sediments. We now clarified it in the manuscript and please see lines 141-142.

Line 155: Consider a different word choice than catastrophic here and elsewhere in the paper.

Done. The word “catastrophic” is changed to “strong” and “high-energy” in the paper. Please see lines 149 and 161.

Line 161: How old were the oldest HMW-FA? This needs to be stated to follow the meaning of this sentence.

Done. The pre-depositional ages of HMW-FA have been indicated in lines 167-168.

Line 167: How is the composition expected to be different that it would result in a different end member age?

Thanks for pointing this out. We suggest a different OC_{terr-bio} composition from a perspective of the organic carbon age. In our study, the OC_{terr-bio} is defined as a mixture of various sources. It does not only represent “fresh” OC but also pre-aged OC_{terr-bio} that could be derived from permafrost deposits or deeper soil profiles, or even highly-degraded OC_{terr-bio}. During the last

deglaciation, the LIS has restricted vegetation development, resulting in fewer contributions from young/fresh OC. Therefore, we propose that the mean age of $OC_{terr-bio}$ should be much older than that of the contemporary system. This is why we chose to use HMW-FA age as endmember values instead of a fixed endmember value from the contemporary system. We added an explanation for this. Please see lines 170-173.

Line 176-178: The “delay” of an event may also result from lateral transport of sediment OC. Could this process affect the dating of this or other events in this study?

We believe this “delay” is more likely induced by a less-constrained chronology. The nearby core JPC15/27 with a recovery of 13 m has documented high sedimentation rates and laminated sediments between 13.5-14.4 kyr BP (about 6-12 m in the core)¹². We have carefully compared the chronology of core ARA04C/37 (recovery of about 6 m) and JPC15/27 in Wu et al. (2020)¹³, and the radiocarbon dates of these two cores are consistent during the last deglaciation. If we include the AMS14C dates from the lower part of core JPC15/37, the first event in our core may have an older age. Therefore, we believe the delay is rather an “apparent delay” and can be attributed to a less-constrained chronology or that our archive has only documented part of the event. We added discussions. Please see lines 185-188.

Line 179 “Although the distinct increases in TOC MAR were linked...”: add “likely” or similar. I suggest to use more careful phrasing here and elsewhere in the paper.

Done. Please see, for example, line 189.

Line 180: What parameters of bulk OC? Looking at the 14C of bulk OC I see very large changes (i.e. a drop of the pre-depositional age from ~30 to ~20 kyr). How do you explain this difference if its not a change in source or composition?

Thanks for this comment. We indicate the parameters clearer, i.e., $\delta^{13}C$, CPI, $f\beta\beta$, and T_{max} (Fig. 2). Please see line 190.

The decrease in pre-depositional age from 30 to 20 kyrs is probably attributed to vegetation development since the LIS retreated. Vegetation development may result in a younger mean age of $OC_{terr-bio}$ (please, compare to “Reply Line 167”), but it does not necessarily mean a larger contribution from $OC_{terr-bio}$ to the sedimentary OC. The younger $OC_{terr-bio}$ (rich in $\Delta^{14}C$) may cause a younger apparent age of bulk OC. This is also supported by our mixing model output which shows decreasing bulk OC ages while OC_{petro} dominated during the last deglaciation (Fig 4).

Line 213 and 219-222: Wouldn't also the CO₂ drawdown by chemical weathering increase? As described above, I think this needs to be included in this discussion to reflect on the net CO₂ exchange with the atmosphere.

The CO₂ drawdown by silicate weathering may also increase during the deglaciation. In the manuscript, we focus more on the OC dynamics and its climate impact. At the end of the manuscript, we added discussions about other processes and emphasized the importance of a comprehensive understanding of the net CO₂ budget. Please see lines 343-370.

Line 234-236: We can or we may expect? This and other sentences of this paragraph are rather speculative and needs to be more carefully phrased.

Done. The sentences are carefully phrased. Please see lines 251-259.

Line 236: Sediments reactive surface areas are also considered to sequester OC to form mineral-organic associations, which stabilize and protect OC from remineralization. I would argue that this process rather limits the reactivity of the OC.

As pointed out by Reviewer#3, OC_{petro} in many cases probably already comes associated with lithogenic minerals. Therefore, our previous argument that mineral protection may cause higher burial efficiency of OC_{petro} during the last deglaciation is not correct. We deleted this argument.

The effect of organic matter protection on mineral surfaces that the reviewer mentioned here is likely more important for fresh biospheric organic carbon, like freshly produced phytoplankton debris that attaches to the mineral surface is protected from oxidation through this adsorption (or its entrainment in pores).

Line 243-252: What area (km²) was used here? Are glaciated areas considered to produce CO₂ as well? Was the area allowed to change through time with regard to glacier retreat? More background information is needed to follow this calculation.

We thank the reviewer for this comment and we improved our description of the approach. We assume that outcropping shales that were previously covered by ice sheets would start oxidation upon exposure to the atmosphere. Therefore, the A_{exposure} indicates the areas with outcropping shales that were freshly exposed during the ice-sheet retreat. We emphasize “freshly exposed area” because we have considered dynamic changes in both the exposed area and the oxidation flux over time. We assume that the freshly exposed shales generated a higher oxidation flux and this flux decreases over time. Therefore, the A_{exposure} is calculated based on the shales distribution from Amiotte Suchet et al. (2003)¹⁴ and the changes in ice-sheet extent from Peltier et al. (2015)¹⁵ and is calculated for every 500 years to indicate the freshly exposed areas in different time periods. The results are shown in Figure 5a. Please see the revised text in lines 260-273.

Areas that remained glaciated are not considered to produce CO₂, only when the areas with shale outcrops were exposed to the atmosphere do we start to calculate CO₂ release from these areas.

Line 256: What is the flux estimate for 10-yr old shales is based on? This number is 20 times higher than the measured/estimated CO₂ flux in this region. If this is only assumed I think an alternative scenario needs to be presented, in which the age of the substrate has no effect to the oxidation rate/CO₂ production. This may also replace the uncertainty calculations in line 257-259.

The flux estimate is based on an assumption that the weathering rate of OC_{petro} decreases with substrate aging, following the equation $F_{\text{oxidation}} = F_0 \times t^{-0.71}$ ^{16,17}. F_0 is the oxidation flux of freshly produced substrate (10-yr old), and $F_{\text{oxidation}}$ denotes the oxidation flux of substrate at age t . To calculate F_0 , we need a known $F_{\text{oxidation}}$ with a known substrate age t . Here we use the

oxidation flux of $1 \text{ tC km}^{-2} \text{ yr}^{-1}$ which is based on the estimate from Horan et al. (2019)¹⁸ ($0.89 \pm 0.32 \text{ tC km}^{-2} \text{ yr}^{-1}$) in the modern shales-dominated region. This region was exposed between 10-15 cal. kyr BP¹⁹, and thus we assume a substrate age of 10 ± 5 kyrs. We are then able to estimate the flux for 10-yr old shales (F_0) based on the modern oxidation flux and assumed substrate age. We revised the text to make it clearer. Please see lines 273-293.

Line 262: Is this 84 Pg budget based on entire North America or only the Mackenzie river catchment? This does not become clear in this paragraph or elsewhere in the manuscript.

We assume the OC_{petro} oxidation flux from the shales-dominated regions in the Mackenzie River catchment is applicable to entire North America, and thus the budget is calculated for entire North America. We added more information and please see lines 268-295.

Line 304: What are these figures of released permafrost OC based on and why are they divided as such portions between the different warming periods? Is this based on the inundated shelf area and permafrost OC release by coastal erosion in the Arctic Ocean? It appears that most of the Arctic Ocean shelf area was flooded after 11.7 kyr BP (e.g. for the Laptev Sea: Bauch et al., 2001).

Yes, the permafrost carbon release is based on the inundated shelf area in the Arctic Ocean. The global sea level rose by 80 m between 18 and 11 cal. kyr BP. We adopted the approach described in Winterfeld et al. (2018)²⁰ to estimate the permafrost carbon flux. Considering the regional bathymetry, Winterfeld et al. (2018)²⁰ used approaches similar to Köhler et al. (2014)²¹ and Brosius et al. (2012)²² and calculated that about 50% of the Arctic shelf area was flooded during this time period. As the total Yedoma domain has been estimated to lose 259 PgC during the transition between the LGM and the Holocene, the authors took a similar fraction, i.e., 129.5 PgC of permafrost carbon was lost between 18 and 11 cal. kyr BP. Furthermore, Vonk et al. (2012) estimated that 66% of the eroded material was oxidized to CO_2 and released into the atmosphere. The authors thus assumed a total amount of 85 PgC (66% of eroded permafrost carbon, i.e., 129.5 PgC) was released as CO_2 between 18-11 cal. kyr BP.

The portions divided for different periods are approximately scaled to the amplitudes of the biomarker MARs records of these three events in Winterfeld et al. (2018)²⁰. Our modeling exercise is updated relative to the published results in that a new version of the carbon cycle model was used, and that the age of permafrost carbon is set to 10 kyr, which is more realistic in light of our new findings. More importantly, in our exercise, we consider the combined effects of permafrost carbon mobilization and degradation and oxidation of petrogenic OC.

Reply to Reviewer #2:

2.1 Petrogenic carbon's oxidation and permafrost OC's erosion have previously been proposed as possible mechanisms substituting the ocean ventilating theory in explaining deglacial atmospheric CO_2 fluctuations. This study tried to make sense quantitatively; while reading through the MS, I doubted the calculation, relatedness, and contribution from this study. First of all, the estimate of the amount of OC_{petro} bases heavily on an equation published in a Biogeoscience Discussion, which is still not accepted by Journal Biogeosciences. As a critical

parameter of this equation, the oxidation rate (0.89) was adopted from another study (Horan et al., 2019). However, while reading through this key reference, the only place that mentioned this same oxidation rate was under the discussion of carbonate weathering. Additionally, the calculation of weathering rate is not constant, as deglacial retreat would largely thicken the soil and thus decrease the weathering rate. Overall, I doubt the calculated 84.3PgC release through the OC_{petro} erosion. Furthermore, I do not see the linkage between the budget calculation and the core measurements from this study. This study is on the sedimentary record from the Beaufort Sea, while the budget calculation of OC_{petro} oxidation is purely based on hinterland data published previously. Second, this study further reinforced the importance of permafrost erosion as an essential source of atmospheric CO₂ during the deglacial time. However, I do not see how much this study contributed rather than repeating what was published by Winterfeld et al. (2018). Overall, I doubt the carbon budget calculation and the contribution solely from this study.

Reply 2.1: We thank the reviewer for this comment. Overall, we believe that the reviewer's major concerns about our budget calculation and the relatedness are due to an insufficient description of our approach in the initial version. We made substantial revisions to this part. Please see lines 260-296. Below we reply to each question in this comment.

The equation ($J_{\text{carbon}} = A_{\text{exposure}} \times T_{\text{exposure}} \times F_{\text{oxidation}}$) for budget calculation is valid because the unit of oxidation flux ($\text{tC km}^{-2} \text{yr}^{-1}$) indicates that it requires a multiplication of the area by the time over which the oxidation occurs to calculate the amount of carbon released. Therefore, we deleted the citation of Blattmann (2022), although we also note that the manuscript is now published as a peer-reviewed paper.

The oxidation flux of $0.89 \text{ tC km}^{-2} \text{yr}^{-1}$ is a mean value we take from the main tributaries of the Mackenzie River (Horan et al., 2019)¹⁸ to represent oxidation fluxes in the shales-dominated region. We now indicate in the manuscript that the number is a mean value from Horan et al. (2019). Please see lines 232-234.

We have not clearly described our concept of how we account for decreasing oxidation fluxes with increasing exposure time in the initial version. In the revised version, we explained that the oxidation flux is known to be $0.89 \pm 0.32 \text{ tC km}^{-2} \text{yr}^{-1}$ in the contemporary system¹⁸ and is estimated to be $10.7 \pm 6.6 \text{ tC km}^{-2} \text{yr}^{-1}$ during the last deglaciation. The changes in oxidation fluxes may reflect processes such as soil formation and vegetation development. To include such processes in our calculation, we assume that the oxidation flux decreases with increasing exposure time ($F_{\text{oxidation}} = F_0 \times t^{-0.71}$). Please see our revisions in lines 273-289.

The linkage between budget calculation and core measurements is not clearly described in our initial version. The concept is that today, the shales-dominated regions in the Mackenzie River catchment have a modern OC_{petro} oxidation flux of $0.89 \pm 0.32 \text{ tC km}^{-2} \text{yr}^{-1}$ ¹⁸. Due to the increase in OC_{petro} MAR (12 ± 6 times) in marine sediments and an assumption of constant burial efficiency, we postulate that the past oxidation flux in the Mackenzie River catchment also increases 12 ± 6 times (i.e., $10.7 \pm 6.6 \text{ tC km}^{-2} \text{yr}^{-1}$). Thereby, the linkage is established between the increased OC_{petro} MAR and the past oxidation flux on land. We here assume that OC_{petro} oxidation fluxes in the shales-dominated regions from the Mackenzie River catchment can be

applied to other shales-dominated regions in North America. Furthermore, based on the modern oxidation flux¹⁸ and the estimated past oxidation flux, we simulate the long-term behavior of OC_{petro} oxidation flux that decreases with increasing exposure time, following the equation $F_{\text{oxidation}} = F_0 \times t^{-0.71}$. F_0 is the oxidation flux of freshly exposed substrate (10-year old), and $F_{\text{oxidation}}$ denotes the oxidation flux of the substrate at age t . Simulation of the long-term behavior requires a F_0 and please see the reply above to “Line 256” for how F_0 is derived. Although the long-term behavior of $F_{\text{oxidation}}$ is implemented based on the modern oxidation flux, we compared with our estimated flux that falls well within the equation and stands for a substrate age of <100 to 2000 years (indicated in Fig. 5b), thus to some degree supporting our assumption. Please see our revisions in lines 226-230, 236-238, 274-275, and 290-293.

First of all, our carbon cycle simulation includes both OC_{petro} release during the glacial retreat and permafrost carbon release from coastal erosion, as both processes may have released terr-OC that is related to the ice-sheet retreat. In addition, as explained in the manuscript, previous studies on permafrost carbon remobilization were carried out in largely unglaciated regions and it is difficult to unambiguously determine the process of coastal erosion. It means that the carbon release proposed by Winterfeld et al. (2018)²⁰ is tentative. Our study confirms that coastal erosion during the rapid sea-level rise was a major process to remobilize permafrost carbon. This finding corroborates the scenario of pulsed carbon release as found in Winterfeld et al. (2018)²⁰ and convinces us to include this process in our simulation as it is related to ice-sheet retreat. Moreover, we used an updated model scenario compared to Winterfeld et al. (2018)²⁰, including an adjustment of the assumed age of eroded permafrost deposits (changed to 10 kyrs), reflecting the growing database.

2.2 It is always important to consider water depth variation while interpreting data generated from a sediment core offshore. Since the last glacial time, the sea level has increased by ~170m. During the deglacial time, the location where sediment cores were retrieved was much shallower. What matters more is that the sampling locations are closer to the coast. Such deglacial-interglacial differences could largely explain the sedimentation rate differences. Explicitly, the sedimentation rate is expected to be much higher when the sampling location was closer to the coastline during the deglacial time. It essentially explains higher OC_{petro} and OC_{permafrost} MARs at the deglacial time. Additionally, the results and discussion rely heavily on one sediment core, while the other core is only presented with limited data and discussion.

Reply 2.2: We thank the reviewer for this comment and agree that the sea level may influence the distance between the coast and core location, which further influences the sedimentation rate. However, the Beaufort Sea has a much narrower shelf than other Arctic Seas (Fig. 1a). As shown in Figure 1b, the red area indicates the flooded shelf since the Last Glacial Maximum. Based on a rough estimate, the coastline retreat (since the LGM) has increased the distance between the river mouth and core location by ~110 km, which is much less than in the extensively flooded Eurasian continental shelves. The number may be further reduced if we only consider the changes during the last 14 kyrs. We added an argument for higher MARs due to a shallower water depth during the last deglaciation. Please see lines 247-248.

This study focuses more on the core ARA04C/37. In our previous study, we have carefully compared the chronology and magnetic susceptibility between core ARA04C/37 and core

JPC15 (Wu et al., 2020)¹³. The two cores have similar chronology and magnetic susceptibility and documented the same timing of the YD flood. More importantly, there are common bits of black particulate matter found in both cores, which are likely petrogenic. We believe the strong OC_{petro} input was the most important feature of deglacial terrestrial OC remobilization and both cores have documented that. Because the study we have carried out for core ARA04C/37 is of a heavy workload, we thus only analyzed samples of B/A interval from core JPC to extend our records and further support the scenario of strong OC_{petro} input.

2.3 The end-member model and the associated results doubted me. First of all, the $\delta^{13}\text{C}$ was adopted from Hilton et al. (2015), who further cited Johnston et al. (2012). Johnston did report $\delta^{13}\text{C}$ values of sedimentary rocks in this region; unfortunately, all the $\delta^{13}\text{C}$ numbers are from the Neoproterozoic strata, which do not represent the average sedimentary rock $\delta^{13}\text{C}$ values in this region with the widespread presence of Phanerozoic strata. Second, $\Delta^{14}\text{C}$ values vary for terrestrial end-members over time; therefore, the authors picked $\Delta^{14}\text{CFA}$ as $\Delta^{14}\text{C}$ values for soil end-member values. It is commonly perceived that there is an offset between bulk and CSRA $\Delta^{14}\text{C}$ values, and such offsets vary with time, especially in this region. Additionally, all data numbers closely cluster together (Fig. 4b); hence, the error generated from the three end-member-model could be enormous.

Reply 2.3: We appreciate this comment and agree that it is not appropriate to use $\delta^{13}\text{C}$ numbers from the Neoproterozoic strata to represent the sedimentary rock $\delta^{13}\text{C}$ values in this region. In the Mackenzie River catchment, there are three geological units: the North American Cordillera, the Interior Platform, and the Canadian Shield. The North American Cordillera and the central Interior Platform are likely the major contributor of OC_{petro} during glacial retreat. According to the geological map of Canada, these two geological units mainly include rocks of the Cambrian to Cretaceous age. Unfortunately, we could not find proper $\delta^{13}\text{C}$ values from these two units in the Mackenzie River basin. Therefore, we turned to the nearby regions to obtain representative endmember values. We assume that the records from Alberta are representative of sedimentary rocks from the Interior Platform and the records from British Columbia are representative of sedimentary rocks from the Cordillera. Then, we get a mean $\delta^{13}\text{C}$ value of $-29.0 \pm 1.4\text{‰}$. Because these records from the Canadian sedimentary rocks are mainly from Permian to Jurassic. We further test a mean $\delta^{13}\text{C}$ value of $-28.6 \pm 1.3\text{‰}$ of Cambrian to Cretaceous rock that has been calculated from the global compilation. Our results remain the same when using both $\delta^{13}\text{C}$ values. We updated our results in the revised version. Please see our discussions in detail in Supplementary Information.

We admit that HMW FAs may not always be representative of bulk OC_{terr-bio}. Despite the potential limitations, the reasons/advantages of using F14CFA as endmember values have been addressed in “Reply 1.2”. Overall, the reasons are (1) HMW FAs can to some degree represent the dynamics of ice sheets, vegetation development, and permafrost formation, and (2) since distinguishing pre-aged OC_{terr-bio} and OC_{petro} can be difficult in some cases, using endmember values from contemporary surface soil will most likely cause an overestimate of OC_{petro}, and (3) molecular-level signature circumvents the constraint of unknown processes during transport which may change the radiocarbon signals from land to the core location. Therefore, using F14CFA has obvious advantages over its limitations.

2.4 The application of some “classic” parameters. Rock-Eval is commonly applied as a method in estimating the thermal maturity of source rocks. However, Rock-Eval analysis on modern biomass, soil, and sediments generates similar Tmax values. The observed variation of Tmax downcore could fully be explained by the diagenetic alteration over the past 14kyr and is not a robust proxy in demonstrating petrogenic carbon inputs. Besides, $f\beta\beta$ is not valid when you assume the petrogenic carbon is mature or over-mature already since thermal cracking breaks down organic molecules. Additionally, the soil has been found to produce significant amount of $\beta\alpha$ hopenes, rather than $\beta\beta$ hopenes, even when the soil is “fresh”.

Reply 2.4: We gratefully thank for this comment. We may have overinterpreted the variations in the initial version and we agree that there are limitations of these classic parameters. As proposed by many studies^{4,23–25}, the contemporary Mackenzie River system has significant contributions from OC_{petro} and all our parameters are in agreement with this characteristic and do not conflict with each other. Besides, the bulk OC ages and the mixing model outcome all suggest an enhanced OC_{petro} input during the last deglaciation. We, therefore, believe that the high Tmax, low CPI, and low $f\beta\beta$ during the last deglaciation are indicative of large OC_{petro} input. In the revised manuscript, we deleted the overinterpretation of data variations and used these classic parameters as supporting evidence for our findings. Please see our revisions in lines 122-133.

Minor comments

Line 57: The above evidence only supports mobilization, but not oxidation of OC_{petro} .

Done. Please compare to the reply to Reviewer#1 (reply to lines 54-57). We rephrased it and please see lines 65-66.

Lines 87-89: thermal maturity based on Rock-Eval, in this study, does not indicate petrogenic OC inputs.

Done. We clarified that we use these parameters as supporting evidence. Please see lines 96-97.

Lines 204: deglacial does not cause physical erosion; glacier advancing does.

The sentence is deleted in the revised version.

Lines 204-209: shorter transport distance is more reasonable than the intense physical erosion of bedrock in explaining enhanced OC_{petro} MAR.

We agree this may be an important factor causing enhanced OC_{petro} MARs. However, the ice extent in Dalton et al. (2020)¹⁹ shows a largely unglaciated Mackenzie River basin since 12 cal. kyr BP (please see figure below), while our source apportionment still indicates the dominance of OC_{petro} input between 12-8 cal. kyr BP. This, we think, is most likely caused by intense physical erosion of bedrock.

Redacted

(Figure source: Dalton et al., 2020¹⁹)

Lines 207-209: The burial efficiency might be higher during the deglacial time since these particles have been deposited along floodplains and have gone through preliminary oxidation.

Here we defined the burial efficiency as the preservation of OC between release on land and long-term burial in marine sediments. Hence, such a process, e.g., preliminary oxidation along floodplains would not increase the burial efficiency under our definition. We added the definition of burial efficiency in lines 141-142.

Lines 219-240: this section presented a contradictory discussion on whether OC_{petro} is extensively protected or weathered. The first part argues that OC_{petro} is protected due to mineral protection, while the second part claims that the OC_{petro} is heavily weathered due to high surface area.

Thanks for pointing this out. In our initial version, we tended to discuss on the molecular level the effect of organic matter protection on the mineral surface. However, this may not be the case for OC_{petro}, we, therefore, deleted this argument. Please compare to reply to Reviewer#1 (reply to line 236).

Instead, we argue for increased OC_{petro} oxidation due to the high surface area. Rock fragments that already contain organic matter as an intrinsic part of the entire matrix will allow more access (of oxygen or microbes or both) to the organic matter the larger the surface area is relative to the mass. So smaller ground rock fragments offer more access to organic matter than larger fragments.

Lines 239-258: The discussion of the budget is based on several criteria, which essentially is not related to the core of this study, which is what generated from the sediment cores.

Please compare our reply to comment 2.1.

Lines 241: Blattmann (2021) was not accepted and should not serve as an essential reference here. Additionally, I do not see this equation being presented in Blattmann (2021, biogeoscience discussion).

Please compare our reply to comment 2.1.

Line 251: This is not the number from the original study (Horan et al., 2019).

We stated that this is not the number from the original study. Please see lines 232-234.

Lines 259-273: the discussion of permafrost relies heavily on Winterfeld et al. (2018); hence it is hard to extract the new contribution from this study.

Please compare our reply to comment 2.1. Besides, we revised this paragraph due to some repeats. Please see lines 297-302.

Line 268-270: residual permafrost/soil on the floodplain or what has been deposited at the coastal region could supply aged permafrost carbon to the study area. This permafrost has been exposed to the atmosphere or has gone through fluvial transport and hence is more resistant to degradation.

Thanks for this comment. We added the possible carbon source (residual permafrost/soil in the coastal region) to the discussion. See lines 301-302.

Reply to Reviewer #3:

3.1 The Results section and Discussion section should either be strictly separated or merged. Currently, considerable discussion resides in the results part (e.g., 132-141, and many other bits and pieces) and leads to some repeats in the manuscript. In conjunction with this, the permafrost vs. OC_{petro} components of the interpretation are presented in a confusing manner and I got the impression that the authors were unsure about their own interpretation. Regarding the presentation of the results, if the authors decide to merge their results and discussion, I suggest that the authors use the bulk radiocarbon data to build their core story and use other datasets (e.g., dated compounds, GDGTs, etc.) as complementary components to build the discussion. The bulk radiocarbon data seem to provide a clear interpretation around which the complementary datasets provide interesting discussion avenues.

Reply 3.1: Thanks for the suggestion. We deleted repeats (e.g., lines 140-143, 297-302) while keeping the Results section and Discussion section separated. Overall, the $14C$ - OC age and other parameters on bulk-level point to a most important feature during the last deglaciation – the substantial remobilization of OC_{petro} , whereas the compound-specific radiocarbon dating of long-chain FAs, which indicate a more specific source of terrestrial biospheric carbon, reveals the occurrences of rapid coastal erosion. We believe our compound records provide strong evidence for coastal erosion, which contributes to a long-term debate on mechanisms of permafrost carbon release and is important for estimating permafrost carbon release from the shelves, and thus we describe the results in detail in a separate section.

3.2 The definition of terrestrial biosphere and distinguishing or equating that with permafrost carbon is unclear. For example, in lines 173-174 the HMW-FAs were interpreted as stemming from permafrost and immature OC_{petro} . However, in the source apportionment in lines 452-454, HMW-FAs ages are used to represent the age of terrestrial biospheric organic carbon. So is

terrestrial biospheric organic carbon equated with permafrost carbon? And in this case, is OC_{petro} also considered terrestrial biospheric carbon of ancient origin? Overall, the usage of the terms “ancient”, “terrestrial biospheric organic carbon”, “permafrost”, etc. lead to some confusion. I recommend the authors refine the usage of their terminology to make the interpretation clearer.

Reply 3.2: The terrestrial OC can be classified into two categories, i.e., OC_{petro} derived from rocks and $OC_{\text{terr-bio}}$ derived from a variety of organisms. The latter includes OC from riverine and lacustrine primary production, plants, and soils. $OC_{\text{terr-bio}}$ derived from riverine and lacustrine primary production is typically highly bioavailable and rapidly degraded resulting in negligible exports to the marine sediments²⁶. More refractory $OC_{\text{terr-bio}}$ compounds derive from soils and plants and are common constituents in marine sediments, and we use long-chain FAs to represent this major $OC_{\text{terr-bio}}$ source. Permafrost soil is a unique and often carbon-rich feature of soil deposits in the high latitude, and we regard permafrost carbon as one of the components of $OC_{\text{terr-bio}}$ which may contribute a lot to this region. However, because permafrost deposits can only form in unglaciated regions, it is not so clear whether all deposits along the Beaufort coasts are permafrost. For instance, Dyke et al. (2004)²⁷ showed a map of LIS extent in which the coasts were less glaciated, while in an updated map, Dalton et al. (2020)¹⁹ proposed that the LIS extended to the continental shelf (east of the Mackenzie River) (Figure 1). If this is the case, soil carbon from the western coasts (e.g., the Yukon coast) is likely the source of permafrost carbon, but we cannot call the soil carbon from the eastern coasts “permafrost carbon”. Therefore, in Line 173-174 (initial version), we explained that the old $OC_{\text{terr-bio}}$ could be derived from coastal permafrost deposits (e.g., western side) and/or ancient OC-rich deposits (e.g., eastern side).

The term “ancient” in this study only indicates the old carbon age. Usually, $OC_{\text{terr-bio}}$ found in marine sediments is pre-aged. However, the range for “pre-aged” is quite large and some $OC_{\text{terr-bio}}$ is very old in our case, we thus use the term “ancient carbon” to indicate both the very old $OC_{\text{terr-bio}}$ and radiocarbon-free OC_{petro} . In Line 95-96 (revised version), we indicated that “ancient carbon may consist of pre-aged terrestrial biospheric organic carbon and radiocarbon-free OC_{petro} ”.

3.3 How was the Monte Carlo simulation implemented? How was the uncertainty of the marine reservoir age considered? Seems like two ages were considered, but how was this implemented numerically? Was the uncertainty of the age model of the core also incorporated in the calculation of uncertainty for the different pools of OC? Given the uncertainty of the age model at the base of the core(s), this seems warranted.

Reply 3.3: We appreciate this comment. In the revised version, we used $F^{14}C$ instead of $\Delta^{14}C$ to implement the Monte Carlo simulation and discussed the endmember values and robustness of the mixing model in detail in the Supplementary Discussion.

Briefly, because $F^{14}C_{\text{marine-bio}}$ depends largely on the radiocarbon content of DIC in surface waters, $F^{14}C_{\text{marine-bio}}$ is thus assumed the same as $F^{14}C_{\text{surface}}$ (fraction modern carbon of DIC in surface water). As marine reservoir age can be expressed as $R = -8033 * \ln(F^{14}C_{\text{surface}}/F^{14}C_{\text{atm}})$ ²⁸, we derive $F^{14}C_{\text{marine-bio}}$ ($F^{14}C_{\text{surface}}$) based on the regional R and $F^{14}C_{\text{atm}}$ (based on IntCal13).

Note that $F^{14}C_{\text{atm}}$ varies through time, therefore we used variable $F^{14}C_{\text{marine-bio}}$ in the mixing model. To test the model sensitivity, we tested $F^{14}C_{\text{marine-bio}}$ values derived from scenarios of $R=405$ years and 1000 years respectively. We assumed an uncertainty of 100 years for each R (to keep consistency with the uncertainty used in the age models of Wu et al. 2020¹³ and Keigwin et al. 2018¹²), and then derived an uncertainty of $F^{14}C_{\text{marine-bio}}$.

We tested our model sensitivity by incorporating the uncertainties of the age model, which has negligible impacts on our mixing model outcome. Please see more information about the mixing model, endmembers, endmember values, and robustness we added in Supplementary Discussion.

3.4 What is the uncertainty/reproducibility of the Rock-Eval results for T_{max} ? For a non-expert, the approximately 12°C spread in T_{max} (Fig. 2g) seems narrow and a discussion of the uncertainty appears warranted.

Reply 3.4: We thank for this comment and agree that we may have overinterpreted the T_{max} variations of approximately 12°C in the initial version, which is relatively narrow. In the revised version, we used T_{max} and other parameters to support the evidence for overall strong OC_{petro} input in this region, while avoiding further interpretation of their variations. In this study, evaluating changes in OC_{petro} contribution is achieved by using the dual carbon isotope mixing model. Please compare the reply to 2.4 and see our revisions in lines 96-97 and 122-133.

3.5 Please include (representative) chromatograms for fatty acids, alkanes, etc. for documentation purposes in the supplementary information. Additionally, I felt like the visual display of total mass accumulation rates and total organic carbon content was missing for interested readers to get a feel for the data. I think these data would reside well in the supplementary information. Furthermore, I imagine that a graphical display of mass accumulation rate of the different types of TOC with the resolution made possible by the source partitioning presented in Fig. 4 would also be a valuable addition to the supplemental and could be helpful in aiding the discussion.

Reply 3.5: We thank for this comment. We have included representative chromatograms for FAs and alkanes in Supplementary Figures 7-8. The TOC content and total mass accumulation rates of core ARA04C/37 have already been published in Wu et al. (2020)¹³. However, since we have included two more $AMS^{14}C$ dates in this study, we showed updated total MARs in Supplementary Figure 6.

We also included the graphical displays of MARs of different types of organic carbon in Supplementary Figure 5. Due to the three strong events (i.e., the YD flood event and two more coastal erosion events), defining the TOC MARs that are used to calculate different OC MARs requires caution. We discussed three different ways to define TOC MARs between 14-10 cal. kyr BP (when the three events occurred). The TOC MARs that we used to calculate OC_{petro} MARs in the Main Text is a conservative estimate. Please see more information in the Supplementary Discussion.

3.6 The quantitative estimates of carbon release from permafrost appears to solely come an Asian-based study (Winterfeld et al., 2018) and OC_{petro} oxidation presented largely hinges on assuming a constant burial efficiency and assuming that the sedimentary records are representative for the integrated catchment (sediment dispersal or sorting seems unaddressed). The restricted occurrence of permafrost in the Mackenzie Basin is mentioned (e.g., lines 171, 273), so is the permafrost-released CO₂ from this area considered negligible? Additionally, OC_{petro} mass accumulation rates from the site investigated is claimed to represent deglaciated North America (lines 310-311), which misses inputs from the Alaskan fjords and Canadian west coast (Cui et al., 2016) and from the catchments draining into the Hudson Bay, which also drain OC_{petro}-rich areas of the Western Canadian Sedimentary Basin, as well as moraines (line 241) and glacial lakes (Blattmann et al., 2019) which trap OC_{petro} in these intermediate sinks. Skeptical readers may have a hard time with the patchwork carbon budget and buying into the idea of the constant burial efficiency, but I think there is an opportunity to present this as a conservative estimate.

Reply 3.6: We thank the reviewer for pointing out that the way we present the quantitative estimate of carbon release may not be sufficiently clear. It is correct that we refer to a previous study based on a core from the East Asian continental margin when estimating carbon released from permafrost. However, in the study by Winterfeld et al. (2018)²⁰, the assumption is made that the core record is representative of a large-scale process active across the entire area that lost most of its permafrost cover during the last deglaciation, namely sea-level rise induced mobilization of permafrost deposits that had accumulated on areas that were exposed during the glacial and were flooded during the deglaciation. Making this assumption, Winterfeld et al. (2018)²⁰ use published estimates of the carbon contained in these flooded areas (105°E-128°W; please see figure below), as well as estimates of the fraction of this organic matter that is re-mineralized, to arrive at a quantitative estimate of the effect that permafrost mobilization might have had on atmospheric CO₂. This assumption seems to be justified, as several records published after this study reported similar periods of increased permafrost carbon mobilization, and we also find comparable deglacial OC MAR maxima that we relate to sea-level rise.

In our study, we take a similar approach to derive a quantitative estimate of C release from exhumed petrogenic carbon. We postulate that our core record is representative of the process of petrogenic carbon mobilization in response to the deglaciation of North America. The variation in OC_{petro} flux is thus taken to reflect erosion/exposure of OC_{petro}, much of which might remain on land and could be oxidized following erosion. We then continue by estimating the amount of petrogenic carbon that was exposed and might have been eroded in the regions experiencing glacial ice retreat based on reconstructions of ice margins through the deglaciation combined with geological maps of outcropping shales in these regions (Figure 5a). We then, using modern observations on OC_{petro} oxidation fluxes and assumptions on decreasing oxidation rates with increasing exposure time (Figure 5b), estimate OC_{petro} oxidation flux across the exposed area (Figure 5c). Note that for this approach we took to estimate oxidation fluxes, we consider mainly processes occurring on land and take our core record simply as an indicator of the temporal increase in OC_{petro} exhumation on land. It is therefore irrelevant by which pathway the (likely rather small) fraction of exhumed OC_{petro} that accumulated in the ocean was transported. We thus do not need to consider fluxes to Alaskan fjords or the

Canadian west coast. Please compare our reply to comment 2.1 and see revisions in lines 226-230, 236-238, 255-256, 274-275, and 290-293.

(Figure source: Brosius et al. 2012²²)

Minor comments

The terms remobilization and oxidation are sometimes used imprecisely. For example, in line 27 remobilization is used. However, remobilization per se is a carbon neutral process and in its context a positive climate feedback is proposed. There are a few such instances in the manuscript. Please refine the usage of these terms.

Done. Please see line 29-30.

There are several points of discussion surrounding the mode of permafrost mobilization/decay, including coastal erosion, shelf flooding, and hinterland thawing. However, from a carbon budget perspective, it is unclear why distinguishing between these different modes of permafrost mobilization and decay is important. I think adding reasons or motivation behind answering these questions would be valuable for the readers.

Done. Please see lines 78-81.

Line 36: Anomaly of the atmosphere or ocean? Please specify.

Done. Please see line 39.

Line 135: In this context, how relevant is isostatic uplift over this timescale?

Walcott (1972)²⁹ collected extensive field-based evidence from eastern North America and estimated that “in the center of the uplifted region, the ground has risen 138 meters during the last 6000 years and is rising at a rate of 2 ± 0.5 cm/yr today; there may be another 300 ± 120 meters of vertical motion left before isostatic equilibrium is reached”. These estimates are primarily for paraglacial periods, whereas the pattern of glacio-isostatic response in deglaciation time differs. The author inferred for the deglaciation period that “where the rate of the retreat was rapid, very rapid vertical movements would have occurred”.

Similarly, the study of isostatic effects of the Cordilleran ice sheet (also important for our budget) in British Columbia summarized for the deglaciation that “ (1) In the Courtenay area on eastern Vancouver Island, sea level fell from 150 m a.s.l. to < 21m in a few hundred years, and (2) At Parksville, the sea fell from 108 m to 52 m in a few hundred years” (cf., ref³⁰). The rate of isostatic uplift then decreased exponentially during the latest Pleistocene and early Holocene.

Therefore, rapid isostatic uplift can be expected during the deglaciation.

Line 147: What kind of fieldwork evidence is this referring to? Please specify

Done. Please see lines 150-152.

Line 149: Delete “yet”. This word implies an expectation that may or may not agree with reality.

Done.

Line 151: What sort of baseline? Recorded in what? Please clarify or edit.

Done. Please see lines 155-157.

Line 230, 231: Comment: OC_{petro} in many cases probably already comes associated with lithogenic minerals (e.g., mica and chlorite, Blattmann et al., 2019, Blattmann & Liu 2021).

We thank for this comment. We deleted our argument for the effect of mineral protection of OC_{petro}.

Line 262: I suggest reducing the number of “significant” digits to 84 ± 30 PgC.

Done. We reduced the number of digits in these terms.

Lines 298-304: This sentence needs to be fixed.

Done. Please see lines 326-331.

Line 330: I suggest deleting “Unfortunately”.

Done.

Line 452: Concerning the notation: I suggest using separate notation for age-corrected ^{14}C concentrations, as I assume you are using age-corrected ^{14}C values for assessing the end members?

We agree and thank the reviewer for this comment. We use $F^{14}\text{C}_{\text{bulk-ini}}$ to represent the sample’s fraction modern carbon before deposition. The $F^{14}\text{C}_{\text{terr-bio}}$ (endmember $F^{14}\text{C}$ values of terrestrial

biospheric carbon) derived from HMW-FAs has been age-corrected as well, which we stated in the text. Please see the revisions in Supplementary Information.

Lines 131, 473: Reference 32 seems like a weak as this study looked at a specific time in the Neoproterozoic. Large swaths of the catchment are Cretaceous and Devonian (see geological map of Canada). Recovering data hidden in other studies seems worthwhile to get a representative picture on the stable carbon isotope composition of OC_{petro}, or discussing the representativity of the available constraints.

We thank the reviewer for pointing out the weakness of the endmember values. Since Reviewer#2 has similar comments, please see/compare our reply to comment 2.3. We have updated OC_{petro} endmember values and the mixing model outcome in the Main Text. Please see our discussions on OC_{petro} endmember values in Supplementary Information.

Language comments

The singular and plural forms for bedrock are the same: bedrock. See dictionaries Merriam Webster, etc.

Done.

Please refine text for smoother reading. For example:

Line 63: delete “within the deglaciated terrains”

Done.

Line 65: delete “might have” (hypothesis implies this)

Done.

Lines 71, 72: move “during the last deglaciation” to the end of the sentence

Done.

Please review the usage of articles (the, a, an) and possibly have a native speaker proofread.

Done.

Reference

1. Blattmann, T. M. Ideas and perspectives: Emerging contours of a dynamic exogenous kerogen cycle. *Biogeosciences* **19**, 359–373 (2022).
2. Horan, K. *et al.* Mountain glaciation drives rapid oxidation of rock-bound organic carbon. *Science Advances* **3**, e1701107 (2017).
3. Cui, X., Bianchi, T. S., Jaeger, J. M. & Smith, R. W. Biospheric and petrogenic organic carbon flux along southeast Alaska. *Earth and Planetary Science Letters* **452**, 238–246 (2016).
4. Drenzek, N. J., Montluçon, D. B., Yunker, M. B., Macdonald, R. W. & Eglinton, T. I. Constraints on the origin of sedimentary organic carbon in the Beaufort Sea from coupled molecular ¹³C and ¹⁴C measurements. *Marine Chemistry* **103**, 146–162 (2007).
5. Grotheer, H. *et al.* Burial and origin of permafrost derived carbon in the nearshore zone of the southern Canadian Beaufort Sea. *Geophysical Research Letters* **47**, e2019GL085897 (2020).
6. Eglinton, T. I. & Eglinton, G. Molecular proxies for paleoclimatology. *Earth and Planetary Science Letters* **275**, 1–16 (2008).
7. Eglinton, T. I. *et al.* Climate control on terrestrial biospheric carbon turnover. *Proceedings of the National Academy of Sciences* **118**, e2011585118 (2021).
8. Kusch, S. *et al.* Controls on the age of plant waxes in marine sediments – A global synthesis. *Organic Geochemistry* **157**, 104259 (2021).
9. Couture, N. J., Irrgang, A., Pollard, W., Lantuit, H. & Fritz, M. Coastal Erosion of Permafrost Soils Along the Yukon Coastal Plain and Fluxes of Organic Carbon to the Canadian Beaufort Sea. *Journal of Geophysical Research: Biogeosciences* **123**, 406–422 (2018).
10. Batchelor, C. L. *et al.* The configuration of Northern Hemisphere ice sheets through the Quaternary. *Nature Communications* **10**, 1–10 (2019).
11. Hilton, R. G. & West, A. J. Mountains, erosion and the carbon cycle. *Nature Reviews Earth & Environment* **1**, 284–299 (2020).
12. Keigwin, L. D. *et al.* Deglacial floods in the Beaufort Sea preceded Younger Dryas cooling. *Nature Geoscience* **11**, 599–604 (2018).
13. Wu, J. *et al.* Deglacial to Holocene variability in surface water characteristics and major floods in the Beaufort Sea. *Communications Earth & Environment* **1**, 27 (2020).
14. Amiotte Suchet, P., Probst, J.-L. & Ludwig, W. Worldwide distribution of continental rock lithology: Implications for the atmospheric/soil CO₂ uptake by continental weathering and alkalinity river transport to the oceans. *Global Biogeochemical Cycles* **17**, 1038 (2003).
15. Peltier, W. R., Argus, D. F. & Drummond, R. Space geodesy constrains ice age terminal deglaciation: The global ICE-6G_C (VM5a) model. *Journal of Geophysical Research: Solid Earth* **120**, 450–487 (2015).
16. Taylor, A. & Blum, J. D. Relation between soil age and silicate weathering rates determined from the chemical evolution of a glacial chronosequence. *Geology* **23**, 979–982 (1995).
17. Vance, D., Teagle, D. A. H. & Foster, G. L. Variable Quaternary chemical weathering fluxes and imbalances in marine geochemical budgets. *Nature* **458**, 493–496 (2009).

18. Horan, K. *et al.* Carbon dioxide emissions by rock organic carbon oxidation and the net geochemical carbon budget of the Mackenzie River Basin. *American Journal of Science* **319**, 473–499 (2019).
19. Dalton, A. S. *et al.* An updated radiocarbon-based ice margin chronology for the last deglaciation of the North American Ice Sheet Complex. *Quaternary Science Reviews* **234**, 106223 (2020).
20. Winterfeld, M. *et al.* Deglacial mobilization of pre-aged terrestrial carbon from degrading permafrost. *Nature Communications* **9**, 3666 (2018).
21. Köhler, P., Knorr, G. & Bard, E. Permafrost thawing as a possible source of abrupt carbon release at the onset of the Bølling/Allerød. *Nature Communications* **5**, 5520 (2014).
22. Brosius, L. S. *et al.* Using the deuterium isotope composition of permafrost meltwater to constrain thermokarst lake contributions to atmospheric CH₄ during the last deglaciation. *Journal of Geophysical Research: Biogeosciences* **117**, G01022 (2012).
23. Goñi, M. A., Yunker, M. B., Macdonald, R. W. & Eglinton, T. I. The supply and preservation of ancient and modern components of organic carbon in the Canadian Beaufort Shelf of the Arctic Ocean. *Marine Chemistry* **93**, 53–73 (2005).
24. Hilton, R. G. *et al.* Erosion of organic carbon in the Arctic as a geological carbon dioxide sink. *Nature* **524**, 84–87 (2015).
25. Vonk, J. E. *et al.* Spatial variations in geochemical characteristics of the modern Mackenzie Delta sedimentary system. *Geochimica et Cosmochimica Acta* **171**, 100–120 (2015).
26. Wei, B., Mollenhauer, G., Hefter, J., Grotheer, H. & Jia, G. Dispersal and aging of terrigenous organic matter in the Pearl River Estuary and the northern South China Sea Shelf. *Geochimica et Cosmochimica Acta* **282**, 324–339 (2020).
27. Dyke, A. S. An outline of North American deglaciation with emphasis on central and northern Canada. *Developments in Quaternary Science* **2**, 373–424 (2004).
28. Soulet, G., Skinner, L. C., Beaupré, S. R. & Galy, V. A note on reporting of reservoir 14C disequilibria and age offsets. *Radiocarbon* **58**, 205–211 (2016).
29. Walcott, R. I. Late Quaternary Vertical Movements in Eastern North America: Quantitative Evidence of Glacio-Isostatic Rebound. *REVIEWS or GEOPHYSICS AND SPACE PHYSICS* **10**, 849–884 (1972).
30. Clague, J. J. & James, T. S. History and isostatic effects of the last ice sheet in southern British Columbia. *Quaternary Science Reviews* **21**, 71–87 (2002).

Reviewer #2 (Remarks to the Author):

The authors have done tremendously amount of work addressing comments raised by me and other reviewers. After going through the revised MS and the response file, I believe most concerns have been properly resolved, however several comments have not been addressed adequately, which somehow insufficiently supports the conclusion drawn in this study.

1) Mass accumulation rates were calculated to be significantly higher during deglacial time than the Holocene at the coring location, which is partially caused by much closer offshore distance. The sedimentation rates, mass accumulation rates and OC accumulation rates at a location the same offshore distance to the modern case are tremendously lower during the deglacial time than numbers reported in this study. This caused an overestimate of OC accumulation rates at the deglacial time, which is reported to be 12 ± 6 times the modern value and further overestimates the intensity of OCpetro oxidation. Authors admitted that sea level variation is a factor that may have contributed to the overestimation of OCpetro oxidation, however, the upper limit is insufficient in defending the robust story when lacking a proper evaluation. The author further argued that other factors may counterbalance their "over-estimation". One of the factors is the glacial grinding. However, grinding by glaciers, as a mechanism of physical erosion, typically produce coarse grain particles, which is especially true for the case of bedrock erosion. Therefore, the argument that glacial grinding produces sediments with high surface area is invalid. On the other hand, even if finer particles have been generated after glacial erosion, finer particles with higher surface area typically promote OC preservation, rather than oxidation. Overall, the budget calculation is still weakly defended.

2) This study incorporated data and conclusion drawn from Winterfeld et al. (2018) in the modeling part, as a major contribution and the main implication of this study. Unfortunately, I could barely see the novelty by doing so, especially in lines 322-332. Numbers, time periods and respective carbon releases have all been reported in Winterfeld et al. (2018). The authors argued that this current study incorporated both OCpetro and permafrost carbon and confirmed the carbon release from coastal erosion. I am still not convinced by this statement. In lines 181-183, the authors argued that both events co-occurred with the global meltwater pulses and older HMW-FAs. I am afraid this is not robust evidence supporting coastal erosion source in relative to glacier melting-induced hinterland source. Instead, the retreating glacier, through intensified melting, could also promote stronger transport of terrestrial materials. Overall, I am not convinced that the discussion on coastal erosion through HMW-FAs and by citing numbers generated by Winterfeld provided robust evidence demonstrating their points of arguments. To the last, I do admit and agree with authors that the argument and story on OCpetro represent the major contribution of this study.

Reviewer #3 (Remarks to the Author):

Apologies for my delay. Please see PDF.

Review #3 attachment on the following page.

Dear Wu et al.,

The work has improved greatly. Major points:

- 1) In my opinion, the introduction needs refinement to make the case for the study clear. The discussion of permafrost and petrogenic organic carbon (lines 41-76) can be reorganized to better illustrate to the reader the importance and novelty of considering these carbon pools. This is followed by two paragraphs (lines 77-104) that declare the objective/approach of the study which would read much better with some rephrasing/reordering. As a first-time reader of the article, I would probably fail to understand the story/objective/focus that the authors are after.
- 2) The statements regarding HMW-FAs in lines e.g. 165-166 and 182-183 seem to contradict each other. I recommend refining the arguments/wording.
- 3) Please expand on/clarify figure 5b (how it was generated, how it is read) along with lines 238, 277, 292 that are very important for the overall carbon flux calculation.
- 4) Reconsider the significant digits to report (e.g., lines 207, 238, 296).

Minor points:

- 1) Comment on line 213: The “traditional” view of geochemists is that all OCpetro is quantitatively oxidized upon exhumation. See discussion in Hedges (1992). In contrast, organic petrologists and palynologists recognized much earlier that this assumption is untrue, but this knowledge largely failed to permeate into the geochemistry literature. See discussion in Blattmann et al. (2018).
- 2) I recommend superimposing the atmospheric trends of atmospheric CO₂ concentration and radiocarbon isotope composition on figures 5d and 5e to aid discussion and give the reader direct visual reference.

This work reports original research data using highly advanced techniques from a remote and very challenging study location with a novel approach to extracting carbon fluxes from ancient pools potentially of great significance for glacial-interglacial cycles. The Arctic is of great consequence for the global carbon cycle and Wu et al. challenge our basic understanding of it and its changes over glacial-interglacial cycles. The novelty of the approaches along with the disentangling of the natural record raised many questions among the reviewers; however, this also demonstrates that it would also stimulate new discussions among a diverse readership to improve methods/interpretations and challenge the overall approach to studying the carbon cycle during glacial-interglacial times – putting continents much more in the foreground. Future studies will be able to test or refine the findings using strategies that probably still need to be developed; I see this as a forerunner study with potentially much more still to come.

I recommend to the editor for Wu et al. to conduct moderate revisions and seriously consider this article for publication in Nature Communications.

Sincerely,
Thomas Blattmann
01.07.2022 Zurich

References:

- Blattmann, T.M., Letsch, D. and Eglinton, T.I. (2018) On the geological and scientific legacy of petrogenic organic carbon. *American Journal of Science* 318, 861-881.
- Hedges, J.I. (1992) Global biogeochemical cycles: progress and problems. *Marine Chemistry* 39, 67-93.

Author responses to reviewer comments on the manuscript entitled “Deglacial release of petrogenic and permafrost carbon from the Canadian Arctic impacting the carbon cycle” (NCOMMS-21-36247A) by J. Wu, G. Mollenhauer, R. Stein, and co-authors

Reviewer #2 (Remarks to the Author):

The authors have done tremendously amount of work addressing comments raised by me and other reviewers. After going through the revised MS and the response file, I believe most concerns have been properly resolved, however several comments have not been addressed adequately, which somehow insufficiently supports the conclusion drawn in this study.

1) Mass accumulation rates were calculated to be significantly higher during deglacial time than the Holocene at the coring location, which is partially caused by much closer offshore distance. The sedimentation rates, mass accumulation rates and OC accumulation rates at a location the same offshore distance to the modern case are tremendously lower during the deglacial time than numbers reported in this study. This caused an overestimate of OC accumulation rates at the deglacial time, which is reported to be 12 ± 6 times the modern value and further overestimates the intensity of OC_{petro} oxidation. Authors admitted that sea level variation is a factor that may have contributed to the overestimation of OC_{petro} oxidation, however, the upper limit is insufficient in defending the robust story when lacking a proper evaluation. The author further argued that other factors may counterbalance their “over-estimation”. One of the factors is the glacial grinding. However, grinding by glaciers, as a mechanism of physical erosion, typically produce coarse grain particles, which is especially true for the case of bedrock erosion. Therefore, the argument that glacial grinding produces sediments with high surface area is invalid. On the other hand, even if finer particles have been generated after glacial erosion, finer particles with higher surface area typically promote OC preservation, rather than oxidation. Overall, the budget calculation is still weakly defended.

Reply to 2.1: The comment regarding the influence of sea-level change and associated changes in the relative position of the core site and the coast is valid, and we acknowledge that this is one possible control on the mass accumulation rates (see lines 251-252). However, we would like to stress that our budget calculation quantifying the oxidation flux of OC_{petro} (and separating it from the contribution coastal erosion of permafrost carbon makes to aged C fluxes to the sediment) is completely independent of the MAR reconstructions from the sediment core (see lines 266 and following for a detailed description on how the budget was constructed). That means uncertainties in the estimates of past oxidation rates do not affect budget calculation or our conclusion. The changes in MAR are only taken as additional evidence for a change in petrogenic carbon remobilization during the deglaciation. Regardless, we think that the described effect of changes in the offshore position is likely only one of many controls (cf., Stein and Macdonald, 2004, chapter 7)¹ on the MAR and, for the following reasons, may not be decisive for the main findings in our manuscript:

- a) Changes in offshore distance may be an important factor for regions with extensive shelves, e.g., the Laptev Sea and the East Siberian Sea, while for the Beaufort Sea’s narrow shelves, we argue that this influence would be much smaller. We add our argument in lines 252-254.
- b) The factor of MAR increase (6 times) we adopted for this study is a conservative estimate. The adopted average TOC MAR during the last deglaciation is derived from low values in between the three extreme events (i.e., the YD flood event and two coastal erosion events) (Please see Supplementary Discussion text 2 and Supplementary Figure 5-6). If using the average TOC MAR derived from the entire period of 14-10 cal. kyr BP, the factor is more than double of the current estimate.

As for the comment on glacial grinding and OC_{petro} oxidation, we revised our argument in lines 258-259.

2) This study incorporated data and conclusion drawn from Winterfeld et al. (2018) in the modeling part, as a major contribution and the main implication of this study. Unfortunately, I could barely see the novelty by doing so, especially in lines 322-332. Numbers, time periods and respective carbon releases have all been reported in Winterfeld et al. (2018). The authors argued that this current study incorporated both OC_{petro} and permafrost carbon and confirmed the carbon release from coastal erosion. I am still not convinced by this statement. In lines 181-183, the authors argued that both events co-occurred with the global meltwater pulses and older HMW-FAs. I am afraid this is not robust evidence supporting coastal erosion source in relative to glacier melting-induced hinterland source. Instead, the retreating glacier, through intensified melting, could also promote stronger transport of terrestrial materials. Overall, I am not convinced that the discussion on coastal erosion through HMW-FAs and by citing numbers generated by Winterfeld provided robust evidence demonstrating their points of arguments. To the last, I do admit and agree with authors that the argument and story on OC_{petro} represent the major contribution of this study.

Reply 2.2: We respectfully disagree with this comment. With regard to the process of coastal erosion, existing evidence usually comes from unglaciated regions which to some degree cannot confirm the occurrence of coastal erosion (elaborated in lines 78-85). In this context, the novelty and major contribution of our study is the finding of enhanced input of pre-aged terrestrial biospheric OC during MWP in the glaciated regions, which most likely indicates the process of coastal erosion (reasons elaborated in lines 304-307). Since we regard this as great evidence of permafrost carbon release through coastal erosion during the glacial retreat, it is necessary to show the audience how this process individually or combined with OC_{petro} weathering contributed to the CO_2 rise and thus we incorporate the estimate from Winterfeld et al. (2018)². Besides, we adjusted the assumed age of eroded permafrost deposits to 10 kyr in the current model scenario. Nevertheless, the estimate from Winterfeld et al. (2018)² does not add novelty to our study, but our compound-specific radiocarbon record does.

We argue that our compound-specific radiocarbon record most likely indicates carbon sources from coastal erosion rather than the inland material, due to the following reasons:

- a) The Younger Dryas flood event is a good example of inland material delivery. The flood has originated from the hinterland and drained through the Mackenzie outlet to the Arctic Ocean (Keigwin et al. 2018; Wu et al., 2020)^{3,4}. During this period, the carbon age of HMW-FAs became younger (Figure 3e), implying a younger hinterland $OC_{\text{terr-bio}}$. In contrast, carbon age during the MWP 1a and 1b is much older, which indicates different sources.
- b) The events of rapid accumulation during the MWPs seem not related to the glacier-melting-induced hinterland source. None of the records from Keigwin et al. (2018)³ or Wu et al. (2020)⁴ documented enhanced freshening at ca. 14 and 11 cal. kyr BP (discussed in lines 149-161). Therefore, the extreme events of rapid accumulation and largely increased carbon age are most likely a consequence of coastal erosion instead of glacier melting.
- c) Today, this region is one of the most rapid erosion regions in the Arctic. It has a weighted mean coastal erosion rate of 1.12 m yr^{-1} (Lantuit et al., 2012)⁵. Stein and Macdonald (2004)¹ also proposed for this region that “Present coastal erosion, which occurs for most of the coast at rates from $5\text{-}20 \text{ m yr}^{-1}$, must be seen as the continuation of the Holocene marine transgression”, suggesting significant erosion in the past.

We added our discussions in lines 178-183.

Reviewer #3 (Remarks to the Author):

Dear Wu et al.,

The work has improved greatly. **Major points:**

1) In my opinion, the introduction needs refinement to make the case for the study clear. The discussion of permafrost and petrogenic organic carbon (lines 41-76) can be reorganized to better illustrate to the reader the importance and novelty of considering these carbon pools. This is followed by two paragraphs (lines 77-104) that declare the objective/approach of the study which would read much better with some rephrasing/reordering. As a first-time reader of the article, I would probably fail to understand the story/objective/focus that the authors are after.

Reply 3.1: We have reorganized the Introduction part. Please see lines 43-90.

2) The statements regarding HMW-FAs in lines e.g. 165-166 and 182-183 seem to contradict each other. I recommend refining the arguments/wording.

Reply 3.2: We clarified in lines 164-165.

3) Please expand on/clarify figure 5b (how it was generated, how it is read) along with lines 238, 277, 292 that are very important for the overall carbon flux calculation.

Reply 3.3: We expanded in lines 755-759 on how figure 5b was generated and how to read it. The “Time” in figure 5b might be misleading and we replaced it with “substrate age”.

4) Reconsider the significant digits to report (e.g., lines 207, 238, 296).

Reply 3.4: Thanks for pointing this out. We reduced the significant digits for the estimate of the factor of OC_{petro} fraction (line 209), past oxidation rate (line 242, 282, and 297), and carbon budget (lines 301 and 326).

Minor points:

1) Comment on line 213: The “traditional” view of geochemists is that all OC_{petro} is quantitatively oxidized upon exhumation. See discussion in Hedges (1992). In contrast, organic petrologists and palynologists recognized much earlier that this assumption is untrue, but this knowledge largely failed to permeate into the geochemistry literature. See discussion in Blattmann et al. (2018).

Reply: We revised our discussions. Please see lines 215-219.

2) I recommend superimposing the atmospheric trends of atmospheric CO_2 concentration and radiocarbon isotope composition on figures 5d and 5e to aid discussion and give the reader direct visual reference.

Reply: Thanks for the suggestion. In fact, Winterfeld et al. (2018)² have compared the model simulations with the atmospheric trends of CO_2 concentration and radiocarbon isotope composition in their figures 4 and 5.

- a) Overall, the comparison implies that deglacial changes in atmospheric CO_2 and $\Delta^{14}C$ are largely controlled by oceanic processes while carbon from degrading permafrost (85 PgC) contributes much less. Due to the differences in scale, the superimposition of our model output with atmospheric trends in our opinion will not provide much of a visual reference to guide the reader in the discussion.
- b) In the study of Winterfeld et al. (2018)², the authors superimposed the atmospheric trends on model simulations and zoomed in to highlight the potential contributions to abrupt rises, this is

not the focus of our study, in which we examine the overall contribution the combined aged terrestrial carbon sources make to atmospheric CO₂ increase.

Therefore, we decided not to include the atmospheric trends in figure 5. Our study focuses on highlighting the potential source of ancient OC, e.g., permafrost and petrogenic OC, and providing an estimate of the potential magnitude that terrestrial OC processes may influence the atmospheric levels.

This work reports original research data using highly advanced techniques from a remote and very challenging study location with a novel approach to extracting carbon fluxes from ancient pools potentially of great significance for glacial-interglacial cycles. The Arctic is of great consequence for the global carbon cycle and Wu et al. and challenge our basic understanding of it and its changes over glacial-interglacial cycles. The novelty of the approaches along with the disentangling of the natural record raised many questions among the reviewers; however, this also demonstrates that it would also stimulate new discussions among a diverse readership to improve methods/interpretations and challenge the overall approach to studying the carbon cycle during glacial-interglacial times – putting continents much more in the foreground. Future studies will be able to test or refine the findings using strategies that probably still need to be developed; I see this as a forerunner study with potentially much more still to come. I recommend to the editor for Wu et al. to conduct moderate revisions and seriously consider this article for publication in Nature Communications.

Sincerely,

Thomas Blattmann

01.07.2022 Zurich

Reference

1. Stein, R. & Macdonald, R. W. *The organic carbon in the Arctic Ocean*. Springer-Verlag, Berlin (2004).
2. Winterfeld, M. *et al.* Deglacial mobilization of pre-aged terrestrial carbon from degrading permafrost. *Nature Communications* **9**, 3666 (2018).
3. Keigwin, L. D. *et al.* Deglacial floods in the Beaufort Sea preceded Younger Dryas cooling. *Nature Geoscience* **11**, 599–604 (2018).
4. Wu, J. *et al.* Deglacial to Holocene variability in surface water characteristics and major floods in the Beaufort Sea. *Communications Earth & Environment* **1**, 27 (2020).
5. Lantuit, H. *et al.* The Arctic Coastal Dynamics Database: A New Classification Scheme and Statistics on Arctic Permafrost Coastlines. *Estuaries and Coasts* **35**, 383–400 (2012).

Reviewer #2 (Remarks to the Author):

Wu et al. have adequately addressed concerns raised by me and I suggest acceptance of the MS in the current format. --Xingqian

Author responses to reviewer comments on the manuscript entitled “*Deglacial release of petrogenic and permafrost carbon from the Canadian Arctic impacting the carbon cycle*” (NCOMMS-21-36247B) by J. Wu, G. Mollenhauer, R. Stein, and co-authors

Reviewer #2 (Remarks to the Author):

Wu et al. have adequately addressed concerns raised by me and I suggest acceptance of the MS in the current format. –Xingqian

Reply: We thank Reviewer #2 for the review and suggestion.